# Ethnomedicinal documentation, phytochemical characterization, and biological evaluation of the traditional medicinal plants from Swat region of Pakistan

Ajmal Khan[1,2]*, Sujogya Kumar Panda[1,3], Haibo Hu[1,4], Liliane Schoofs[1], Walter Luyten[1]

**1** Department of Biology, Animal Physiology and Neurobiology Section, KU Leuven, Leuven, Belgium, **2** Centre for Animal Sciences and Fisheries, University of Swat, Charbagh Swat, Khyber Paktunkhwa, Pakistan, **3** Centre for Biotechnology, School of Pharmaceutical Sciences, Siksha 'O' Anusandhan (Deemed to be University), Bhubaneswar, India, **4** National Engineering Research Center for Modernization of Traditional Chinese Medicine - Hakka Medical Resources Branch, School of Pharmacy, Gannan Medical University, Ganzhou, China

\* ajmal.khan@kuleuven.be, ajmalkhan399@hotmail.com

## Abstract

Traditional medicinal plants are a primary source of natural products which are used for the prevention and treatment of various infections throughout the world. This study documents the ethnomedicinal investigation, phytochemical characterization, thin layer chromatographic (TLC) profiling and bioactivities of 17 traditionally used medicinal plants, belonging to 12 taxonomic families from the Swat region of Pakistan. The plants were collected after interviewing local ethnomedicinal knowledge holders, and confirmation of their effective use by the local population and available literature. The extracts (85) were prepared in five different solvents (hexane, acetone, ethanol, methanol, and water), and were tested for a range of bioactivities: antibacterial (5 Gram-positive and 9 Gram-negative bacteria), antifungal (6 yeasts), antibiofilm (*S. aureus* and *C. albicans*), and cytotoxicity (cancerous and non-cancerous cell lines). Results demonstrated that 25% of the extracts showed pronounced activity (inhibition value [IV] > 50%) against different planktonic microbes, and 35% against biofilm strains of bacteria and fungi, with ethanol being the best solvent. Cytotoxicity was often observed against a tumor cell, but rarely against non-tumoral cell lines. A number of phytochemical compounds such as alkaloids, flavonoids, phenols, steroids, terpenoids, coumarins, tannins, saponins, chalcones, and quinones were detected in the extracts using standard phytochemical characterization methods, which were further authenticated through TLC separations. This is the first study to report the phytochemical screening, TLC profiling, and bioactivities of these medicinal plants, particularly their antibiofilm properties, which have not been documented previously by other researchers. This work is a significant addition to the field which

**Data availability statement:** All relevant data are within the paper and its Supporting information files.

**Funding:** The author(s) received no specific funding for this work.

**Competing interests:** The authors have declared that no competing interests exist.

reinforces the importance of indigenous knowledge in selecting medicinal plants for drug discovery based on local remedies. In conclusion, plants like *Juglans regia*, *Punica granatum, Artemisia maritima, Aesculus indica, Thymus linearis, Nasturtium officinale, Berberis lyceum, Dysphania ambrosioides,* and *Mentha spicata* show promise for further research as a potential sources for novel drug discovery.

## Introduction

Medicinal plants have been used traditionally throughout large parts of the world for the prevention and treatment of diseases since time immemorial, due to their antimicrobial, antiviral, antioxidant, anticancer, and anti-inflammatory activities, as well as other pharmacological effects of benefit to mankind [1,2]. However, during the 20th century, their usage declined, but interest remained in the scientific basis for their effects. Over the last decades, interest in phytomedicine has rebounded because of the adverse effects of antibiotics and other synthetic compounds on the health of humans and animals, as well as concerns about the quality and safety of products. Since 2006, the EU has also banned the use of antibiotic growth promoters in animal feed, as there is a risk of the transfer of antibiotic resistance to human pathogens. This has increased again the interest in using plants as alternative natural source of medicines, and using their extracts as potential sources of bioactive [3,4].

The traditional local therapeutic use of different plant species has gained more interest, thanks to the confirmation of their therapeutic properties by phytomedicine researchers. Almost 25% of bioactive components of currently prescribed medicines have been identified from medicinal plants [5,6]. According to a report by the World Health Organization (WHO), in developing countries of the world, about 80% of the population is still dependent on traditional plant medicines for their basic health care. The WHO has documented more than 20,000 species of medicinal plants, and considers them as a promising source of novel drugs [7]. Various cultures use traditional medicinal plants due to their preventive, curative, and health-boosting capabilities, including Ayurveda, Siddha, Kampo, Unani, Jamu, traditional Chinese medicine (TCM), Thai herbal medicines, and others [8]. The Rig-Veda, dating from 4500 BC to 1600 BC is one of the oldest records, which provides detailed therapeutic uses of the plants of the Indian subcontinent [9]. Pakistan has several ecological zones with diverse plant ranges offering more than 600 medicinal wild plant species [10]. In Pakistan, more than 1000 species have been identified as medicinal plants, with regular trading of about 350–400 medicinal plant species in different drug markets [11]. Swat is one of the most biodiverse floral districts of Pakistan, with seven different types of forests, ranging from alpine to tropical dry deciduous forests. There are about 55 pteridophytes, 1550 vascular plants, and over 345 traditional medicinal plants which have been described by numerous researchers in extensive ethnobotanical studies [12]. There is a dire need for the preservation of the traditional knowledge about the ethnomedicinal importance of plants for the benefit of future researchers interested in the development of new medicines [10]. The local traditional uses of these plants are

generally in crude extract forms, while new advanced chemical and microbial approaches can lead to the discovery of various potent compounds from the available medicinal plants. At present, many biologically active compounds have already been isolated and identified from traditionally used medicinal plants, but many others remain to be discovered [11]

There is a crucial need for the discovery of new antimicrobial agents due to the increasing resistance to available therapeutic agents for human and animal diseases. There is a need for screening of medicinal plants to pursue their promising bioactivities as there are many plants with purported antiparasitic or antimicrobial properties, which have not been reproduced under clearly defined experimental conditions. Based on the traditional use of the medicinal plants for the treatment of various ailments in Swat, it is hypothesized that the extracts of these selected traditional medicinal plants could be effective for treating various microbial infections. This study opens opportunities for discoveries that could be beneficial in managing various microbial infections. The results can also provide scientific evidence to support local traditional use of select medicinal plants of the Swat region of Pakistan.

## Materials and methods

### Chemicals and reagents

Hexane, acetone, chloroform, and methanol (all HPLC grade) were purchased from Sigma-Aldrich Co. (USA). Absolute ethanol was purchased from Fischer Chemicals (UK). Sterile deionized water was produced by a Milli-Q Reagent Water System (MA, USA. Yeast extract and Bacto™ peptone were purchased from Lab M Ltd. (Lancashire, UK). Dimethyl sulfoxide (DMSO, molecular biology grade), dextrose, sucrose, sodium chloride, ammonium chloride, calcium chloride, ferric chloride, iodine, potassium iodide, sulphuric acid, hydrochloric acid, sodium hydroxide, ammonium hydroxide, magnesium sulfate, potassium hydrogen phosphate, potassium dihydrogen phosphate, di-sodium hydrogen phosphate, cholesterol-ester, TLC plates, antibacterial controls (ciprofloxacin, erythromycin, and chloramphenicol), the antifungal control (miconazole nitrate salt), cytotoxic control (gossypol), fetal bovine serum (FBS), trypsin/EDTA, Dulbecco-Modified Eagle's Medium-high glucose (DMEM), Phosphate-Buffered Saline (PBS), RPMI-MOPS (Rosewell Park Memorial Institute medium with L-glutamine without sodium bicarbonate, and 4-morpholinerpropanesulfonic acid), and penicillin-streptomycin solution (P/S) for cell culture were purchased from Sigma-Aldrich (St. Louis, MO, USA). Resazurin salt was purchased from Acros Organics (Geel, Belgium).

### Study area and design

An extensive field trip was conducted to the Swat region of the Khyber Pakhtunkhwa province of Pakistan for collecting the plant materials. Swat is located (34°34' to 35°55' N latitude and 72°08' to 72°50' E longitude) in the northwest of Pakistan in the range of remote Hindukush mountains. Swat is home to beautiful valleys and snow-capped mountains; therefore it is also known as the Switzerland of Asia. The climate of the region is not very harsh, with a recorded temperature range of 16°C-33°C in summer and 2°C-11°C in winter. The area is typically rainy in summer, with an average annual precipitation of 838.2 mm and snow accumulation of up to 121.92 cm in the winter. It is one of the most biodiverse floral districts of Pakistan, that possesses more than 345 medicinal plants [12] The ethnobotanical studies were carried out as per the guidelines [13]. A total of 17 plants represented in "Fig 1", were selected for this study, and our data were cross-checked by reviewing previous relevant studies. Information on these plants, with their herbarium numbers, scientific name, local name, used part(s), applications, and GPS coordinates is listed in "Table 1."

### Documentation and processing of plants

**Ethical considerations.** The collection of the plant materials were carried out in Swat region of Pakistan. The permission for the research activities was obtained from the Regional Conservator of Forestry, Environment and Wildlife Department Swat after the provision of an authority letter issued by KU Leuven, Belgium. The selection of the plants was

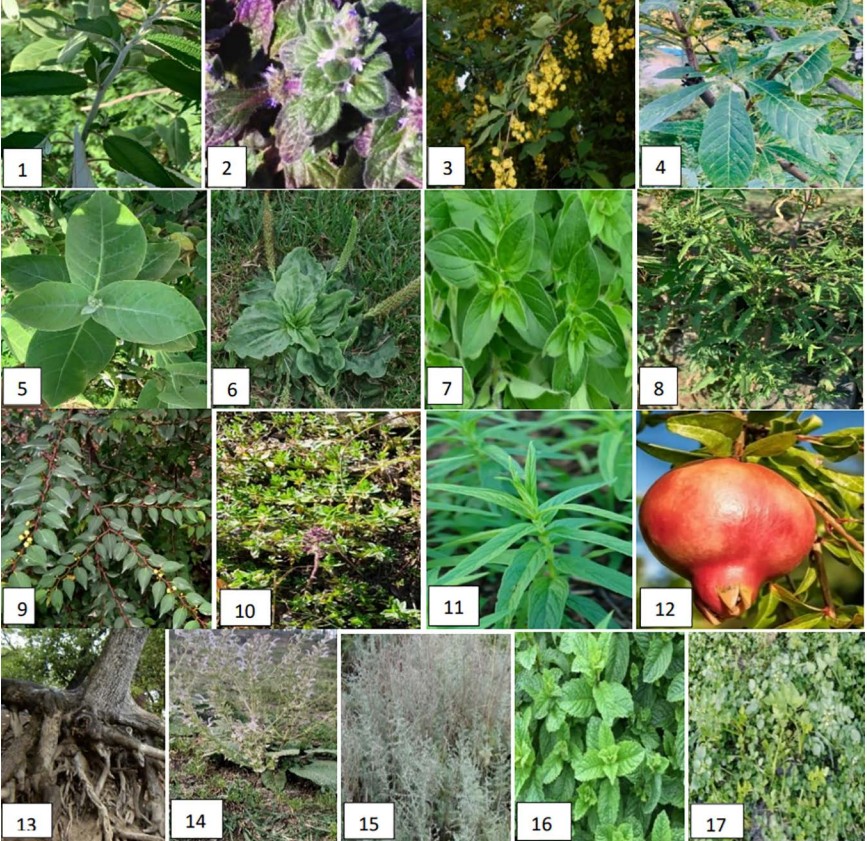

**Fig 1. Photographic representation of 17 studied traditional medicinal plant species collected from Swat region of Pakistan.**

mainly based on their limited phytochemical information and broad-spectrum traditional use. For information collections, the purpose of the study was explained and prior informed consent was obtained from local traditional knowledge holders, and the data were collected in face-to-face interviews as shown in the supporting information ("S1 File").

**Inclusivity in global research.** Additional information regarding the ethical, cultural, and scientific considerations specific to inclusivity in global research is included in the supporting information ("S3 File").

**Botanical identification.** The collected plants were identified at Centre for Plant Science and Biodiversity, University of Swat by following standard authentication methods. The voucher specimens are stored in the Herbarium of the University of Swat, Pakistan.

### Extraction of plant materials

The collected plant parts (whole plant, root, leaf, stem, or bark) were processed for extract preparation by drying in the absence of sunlight for about 1 week at ambient temperature to maintain their green colour and preserve volatile oils [14]. The dried raw plant materials were powdered with an electric grinder, and stored in polyvinyl chloride (PVC) plastic bags in a cold room. Afterward, 1 g of powder from each plant material was transferred into separate sterile 15 mL Falcon tubes, each containing 10 mL of hexane, acetone, ethanol, methanol, or water. The extraction was performed at ambient temperature and enhanced by sonicating 4 times in a sonicator water bath for 15 minutes with a 6-hour interval. After repeated sonication and vortexing, the tubes were centrifuged for 10 minutes at 3500 rpm, and the supernatants were

**Table 1. List of the collected traditional medicinal plants of Swat region of Pakistan.**

| Herb No. | Voucher No. | Latin name | Local name | Taxonomic family | Part used | GPS coordinates and collection site | Traditional use |
|---|---|---|---|---|---|---|---|
| 1 | USP271 | *Debregeasia salicifolia* (D.Don) Rendle | Ajlai | Urticaceae | Leaves | 34˚5725.625'N 72˚2525.625'E (Baidara Swat) | Urinary complaints, skin diseases |
| 2 | USP272 | *Ajuga bracteosa* Wall. ex Benth. | Boti | Labiatae | Leaves | 34˚5713.7N 72˚2325.7E (Sinpora Swat) | Throat infection, fever, skin infections |
| 3 | USP273 | *Berberis lyceum* Royle | Kwaray | Berberidaceae | Root bark | 35˚332.328N 72˚1858.903E (Pashtonai Swat) | Wound healing, ulcers, boils |
| 4 | USP274 | *Aesculus indica* (Wall. ex Cambess.) Hook. | Jawaz | Hippocastanaceae | Leaves | 34˚5725.625'N 72˚2525.625'E (Baidara Swat) | Rheumatism, skin diseases, vein complications |
| 5 | USP275 | *Calotropis procera* (Aiton) Dryand. | Spalmai | Asclepiadaceae | Leaves | 34˚4216.3296N 72˚1338.9748E (Parrai Swat) | Skin diseases, piles, ulcers, tumors |
| 6 | USP276 | *Plantago major* L. | Jabai | Plantaginaceae | Aerial parts | 35˚414.451N 72˚1624.197E (Fazal Banda Swat) | Bronchitis, dysentery, wound healing, burns treatment |
| 7 | USP277 | *Origanum vulgare* L. | Ghat Shamakay | Labiatae | Leaves | 34˚5713.7N 72˚2325.7E (Sinpora Swat) | Diarrhea, skin diseases, respiratory disorders |
| 8 | USP278 | *Dysphania ambrosioides* (L.) Mosayakin & Clemants | Skha Botay | Amaranthaceae | Aerial parts | 35˚332.328N 72˚1858.903E (Pashtonai Swat) | Diarrhoea, dyspepsia, flatulence, colds, coughs, flu, malaria, asthma |
| 9 | USP279 | *Ziziphus oxyphylla* Edgew. | Elanai | Rhamnaceae | Leaves | 34˚5713.7N 72˚2325.7E (Sinpora Swat) | diabetes, fever, skin infections, urinary infections |
| 10 | USP280 | *Thymus linearis* Benth. | Da Ghra Sperkai | Labiatae | Whole plant | 35˚414.451N 72˚1624.197E (Fazal Banda Swat) | Stomach disorders, tonic, jaundice, |
| 11 | USP281 | *Mentha longifolia* (L.) L. | Welanay | Lamiacea | Leaves | 34˚4928.194N 72˚2341.079E (Bara Bandai Swat) | Sore throat, nasal congestion, nausea, flatulence, ulcerative colitis, liver complaints |
| 12 | USP282 | *Punica granatum* L. | Narsaway | Punicaceae | Fruit | 35˚332.328N 72˚1858.903E (Pashtonai Swat) | Diarrhoea, dysentery, whooping cough |
| 13 | USP283 | *Juglans regia* L. | Dindasa | Juglandaceae | Root bark | 34˚5713.7N 72˚2325.7E (Sinpora Swat) | Wounds, dental complaints, acne, chilblains, pharyngitis, diabetes, jaundice |
| 14 | USP284 | *Salvia moorcroftiana* Wall. ex Benth. | Khar Ghwag | Lamiaceae | Leaves | 34˚4928.194N 72˚2341.079E (Bara Bandai Swat) | Skin infections, bronchitis, asthma, cough, itching, dysentery, boils |
| 15 | USP285 | *Artemisia maritima* L. | Tarkha | Asteraceae | Leaves | 34˚4928.194N 72˚2341.079E (Bara Bandai Swat) | Wound healing, ulcers, asthma, intermittent fever, |
| 16 | USP286 | *Mentha spicata* L. | Podina | Lamiaceae | Leaves | 35˚332.328N 72˚1858.903E (Pashtonai Swat) | Stomach and abdominal pain, fevers, headaches, bronchitis, ulcerative colitis, liver complaints |
| 17 | USP287 | *Nasturtium officinale* R.Br. | Thalmera | Brassicaceae | Aerial parts | 35˚332.328N 72˚1858.903E (Pashtonai Swat) | Diabetes, abdominal pain, bronchitis, diuresis, tuberculosis, influenza, asthma |

transferred to 1.5 mL Eppendorf tubes in 1 mL aliquots. The Eppendorf tubes were then subjected to solvent evaporation in a SpeedVac Concentrator, and the dried residue (of 1 mL extract) was weighed. The extracts were re-dissolved at a concentration of 20 mg/mL in DMSO for non-aqueous extracts and in water for aqueous extracts. All the extract samples were stored at 4°C till further use [15].

## Phytochemical screening

The phytochemical screening of the ethanol extracts of select medicinal plants was carried out to determine the presence of secondary metabolites using conventional standard procedures [16–19]. The selection of ethanol extracts was based on their potency in terms of antimicrobial activity and low cytotoxicity as reported in this study. All the extracts were tested ("S2 File") for alkaloids, flavonoids, phenols, steroids, terpenoids, coumarins, tannins, saponins, chalcones, and quinones; which were selected based on their potency in terms of bioactivities.

**Test for alkaloids.** The alkaloids were tested through precipitation reactions using Wagner's reagent test. One mL of the extract was treated with 5 mL of 1% aqueous HCL on a steam bath and the residue was filtered. Afterwards, 1 mL of the filtrate was treated with few drops of Wagner's reagent (a solution of iodine in potassium iodide). The presence of alkaloids was indicated by the formation of reddish-brown precipitate [19].

**Test for flavonoids.** The flavonoids were detected by using alkaline reagent test. In this test, 1 mL of the extract was treated with 2 mL of 2% sodium hydroxide (NaOH) solution, followed by the addition of few drops of diluted hydrochloric acid (HCL). The presence of flavonoids was indicated by the observed color change from intense yellow color to colorless upon the addition of dilute HCL [18].

**Test for phenols.** The phenolic compounds were detected through ferric chloride test. In this test, 1 mL of the extract was treated with 2 mL of 5% ferric chloride ($FeCl_3$). The observation of a blue-green color was taken as the presence of phenols [17].

**Test for steroids.** The presence or absence of steroids was determined by using Salkowski test. In this test, 2 mL of the extract was introduced to 2 mL of chloroform with a subsequent addition of concentrated sulphuric acid ($H_2SO_4$). The appearance of red color in the chloroform layer was regarded as positive test for steroids [16].

**Test for terpenoids.** The terpenoids were tested by using Salkowski test. In this test, 5 mL of the extract was mixed with 2 mL of chloroform and 3 mL of the concentrated sulphuric acid ($H_2SO_4$) was carefully added to form a layer. The positive test for terpenoids was indicated by the presence of a reddish-brown color [19].

**Test for coumarins.** The NaOH test was employed to detect coumarins in the plant extracts. In this test, 2 mL of the extracts was treated with 3 mL of 10% Sodium hydroxide (NaOH). The presence or absence of coumarins was indicated by yellow color [17].

**Test for tannins.** The Braymer's test was used to detect tannins in the plant extracts. In this test, 2 mL of plant extracts was stirred with 3 mL of distilled water, followed by addition of 5 drops of 10% $FeCl_3$. The formation of dark blue precipitate was taken as positive test for tannins [16,18].

**Test for saponins.** The presence or absence of saponins were tested by using foaming test. In this test, 3 mL of plant extracts were vigorously shaken with 3 mL of distilled water. A positive test was indicated by the formation of foam or froth upon shaking [16].

**Test for chalcones.** The presence or absence of chalcones were tested by using ammonia test. In this test, 1 mL of the plant extracts was mixed with 2 mL of 5% ammonia ($NH_3$) solution. A positive test for chalcones was indicated by the formation of a reddish color [19].

**Test for quinones.** The concentrated HCL test was used to detect quinones in the plant extracts. In this test, 1 mL of the plant extracts were mixed with 2 mL of concentrated hydrochloric acid (HCl). The development of green color was taken as positive test for quinones [18].

## Thin Layer Chromatographic (TLC) profiling

The Thin Layer Chromatography (TLC) was performed as per protocol adopted from previous studies [20–22] with some modifications. The crude extracts of the selected medicinal plants were prepared in ethanol at a concentration of 10 mg/mL and were dissolved through vertexing and sonication. The particle free solution (2–5 µL) of the selected extracts were subjected to TLC plates (2.5 × 8 cm) coated with silica gel through a fine bore glass capillary tube over a marked pencil line on lower side of the plate. The spotted plates were kept on the table until complete drying. The sample loaded plates were then placed in a glass chamber containing a mixture of solvents of the chosen mobile phases, i.e., hexane and ethyl acetate (7:3) and methanol and water (4:6), to optimize the separation of various components of the mixture. The developed plates were dried in a fume hood. The dried plates were visualized under ultra-violet (UV) light at a wavelength of 254 nm ("S2 File"). The Rf value of each spot was calculated as:

$$Rf = distance\ travelled\ by\ the\ solute \div distance\ travelled\ by\ the\ solvent$$

## Antimicrobial test

**Microbial strains.** A total of 20 bacterial and fungal strains were used for the antimicrobial testing of the crude extracts of the selected medicinal plants as described [14,23]. The 20 human pathogens (primary/ opportunistic) included 5 Gram-positive bacteria [*Staphylococcus aureus* (ATCC 65385), *Staphylococcus epidermidis* (ATCC 1457), *Streptococcus faecalis* (DPMB 4), *Enterococcus faecalis* (HC-1909–5), and *Micrococcus luteus* (DPMB 3)], 9 Gram-negative bacteria [*Escherichia coli* (ATCC 47076), *Pseudomonas aeruginosa* (PA 01), *Shigella sonnei* (LMG 10473), *Shigella flexneri* (LMG 10472), *Enterobacter aerogenes* (ATCC 13048), *Acinetobacter baumannii* (RUH 134), *Salmonella enteritidis* (ATCC 13076), *Brevundimonas diminuta* (a kind of gift from Prof. Rob Lavigne at KU Leuven) and *Aeromonas hydrophila* (ATCC 7966)], and 6 fungi [*Candida albicans* (SC 5314), *Candida parapsilosis* (ATCC 22019), *Candida glabrata* (ATCC 2001), *Candida auris* (OS 299), *Candida utilis* (IHEM 4005), and *Saccharomyces cerevisiae* (ATCC 7754)]. All the bacterial and fungal strains were maintained at −80°C in a freezer. Before assay, the bacterial frozen stocks were inoculated on Luria-Bertany (LB) agar plates, while YPD agar was used for the inoculation of yeast colonies. The bacterial and fungal plates were incubated overnight at 37°C and 35°C, respectively. The plates were sealed with parafilm and stored in a 4°C refrigerator for future use.

**Preparation of pre-culture.** A single colony of the respective human pathogen was inoculated in separate reaction tubes under aseptic conditions. For *Candida*, 5 mL of YPD medium (1% yeast extract, 2% peptone, and 2% dextrose) and for bacteria, LB medium (1% bacto-tryptone, 0.5% bacto-yeast extract, and 0.5% sodium chloride) was used. The tubes were then incubated for 16–24 hours in a shaker incubator (200 rpm) at 37°C for bacteria and 35°C for yeasts.

**Antibacterial test (Broth microdilution).** The antibacterial test was performed as described in our previous study [24] with some modifications. In short, a 10 µL test sample was added in the wells of a 96-well plate alongside positive controls and solvent controls. Each well was then inoculated with a 190 µL of a standardized microbial inoculum with an optical density (OD) of 0.003 at 620 nm. For control wells, 10 µL extract and 190 µL sterile LB (Lauria-Bertani) broth was added in the concerned wells to correct for any absorption due to extract components. For the positive control, 200 µg/mL (stock) of ciprofloxacin was used, and for the solvent control 5% of DMSO or water was used. The plates were then incubated at 37°C in a shaker-incubator for 24 hours, and were subsequently read on a Multimode Microplate Reader at 620 nm (lamp energy: 13,000) using the MikroWin 2000 software package. The OD values of the wells with a plant extract were corrected for the absorption contributed by the extract. The relative inhibition percentage (%) was calculated by dividing the OD value of the test sample (A) minus that of the non-inoculated extract control (B) by the average OD of the solvent control (C) and multiplying by 100.

*Relative inhibition* (%) = $(A - B \div C) \times 100$

**Antifungal test.** The antifungal activity was assessed using a similar protocol as for bacteria. Instead of LB broth, YPD was used, with a smaller quantity of plant extract (to keep the final DMSO concentration below 2%). For antifungal activity, 4 µL of the test sample was added in each well of the microwell-96 plate, and then 196 µL of the diluted yeast suspension was added to the relevant wells. Control wells were prepared with 4 µL extract and 196 µL YPD broth to correct for any absorption due to extract components, while miconazole (250 µg/mL, stock) was used as the positive control and 2% DMSO/water as the solvent control.

**Determination of IC$_{50}$ (Concentration yielding 50% inhibition).** The IC$_{50}$ was determined using similar protocol as described earlier [24]. The dried extracts were weighed on an analytical balance, dissolved in DMSO or water at a stock concentration of 20 mg/mL. A two-fold serial dilution series (up to 64-fold) was prepared in a V-shaped 96-well plate, followed by antibacterial and antifungal tests as described above. The data of the experiments were presented as % inhibition and analysed with the software package Prism™ (GraphPad Prism 5.0 Software Inc., San Diego, CA). The IC$_{50}$ for each inhibition curve was calculated by non-linear regression.

## Antibiofilm test

The biofilm test was performed by using *Staphylococcus aureus* (USA 300) and *Candida albicans* (SC 5314) according to our previous methods [25] with some modifications. The biofilm strains were grown in TSB medium for *S. aureus* and YPD broth for *C. albicans* at 37°C for 18–24 hours. The tube containing microorganisms was centrifugated at 800 rpm for 2 minutes, and the supernatant discarded. Afterwards, 1 mL of fresh medium was added to the tube and the cells were resuspended by gently vortexing. The OD was measured at 600 nm for each culture, and adjusted to OD 0.1. Subsequently, 100 µL of the *S. aureus* suspension in TSB and *C. albicans* in RPMI-MOPS was transferred into separate wells of a 96-well plate, followed by incubation for 90 minutes at 37°C in a stationary incubator to facilitate the initial adhesion phase of the biofilm formation. After this incubation period, the respective media were carefully removed, and each well was gently washed three times by using 100 µL of PBS. Then the test samples and media were added to each well, consisting of 10 µL of test sample and 190 µL TSB for *Staphylococcus*, or 4 µL of the sample and 196 µL RPMI-MOPS for *Candida*. DMSO or water was used as solvent control while erythromycin (for *S. aureus*) and miconazole (for *C. albicans*) were used as positive controls. Along with solvent and positive controls, one well was kept empty for the resazurin control afterwards during staining. The plates were then kept overnight in a stationary incubator at 37°C. After 24 hours of incubation, the media were removed, and the wells were washed with PBS twice; then, 100 µL of the resazurin dye stock solution (40 µg/mL) was added for staining the cells. After 1 h incubation at 37 °C, the fluorescence was measured with a FlexStation II spectrofluorometer (Molecular Devices, USA) with $\lambda_{ex}$ at 535 nm and $\lambda_{em}$ at 590 nm. The percentage of biofilm inhibition was calculated relative to the growth controls by using the formula:

$$\textit{Biofilm inhibition } (\%) = 100 - (\textit{fluorescence readings of biofilm and samples} - \textit{alamar blue blank control} \div \textit{solvent control} \times 100)$$

Two-fold serial dilutions were tested for determining the BIC$_{50}$ (concentration inhibiting biofilm formation by 50%), which was calculated by non-linear regression using GraphPad Prism 5.0 software (San Diego, CA).

## Cytotoxicity assay

The cytotoxicity test was performed by using resazurin-based cell viability assay with some modifications as described by us previously [8]. The *in vitro* cytotoxicity of the plant extracts was investigated using human lung epithelial tumor cells (A549; obtained from Division of Animal Physiology and Neurobiology, KU Leuven, Belgium) and non-tumoral human lung fibroblast cells (WI-26 VA4; obtained from European Collection of Authenticated Cell Cultures, Sigma Aldrich). The

cell lines were maintained in a humidified 5% $CO_2$ incubator in DMEM supplemented with antibiotics 100 μg/mL penicillin, 100 μg/mL streptomycin, and 10% FBS. For the cytotoxicity test, 200 μL cell culture ($2 \times 10^4$ cells per well) were plated in 96-well microtiter plate and incubated at 37°C. After 24 hours, the media was changed and cells with new DMEM medium were exposed to the plant extracts (20 mg/mL in DMSO or water), DMSO or water (solvent control), and gossypol (positive/cytotoxic control; 10 mM). The plates were again incubated overnight. After 48 hours, 10 μL resazurin solution (0.15 mg/mL in PBS, stock) was added to each well for measuring cell viability. The plates were then incubated for a further 4 hours at 37°C in a 5% $CO_2$ incubator while covered in aluminum foil. The absorbance was measured with a 550nm excitation filter and a 590 nm emission filter in an automated multi-well fluorescence reader (FlexStation II, Molecular Devices, USA). The cytotoxicity was expressed as cell viability inhibition (%), which was calculated from the absorbance values as under.

$$Cell\ viability\ inhibition\ (\%) = 100\% - (treated\ cells - background\ controls) \div (DMSO\ controls - background\ controls) \times 100\%$$

## Data processing and analyses

All bioactivity assays of this study were performed twice for confirmation. The data of all bioactivities of the plant extracts were analyzed by using the webtool ClustVis (https://biit.cs.ut.ee/clustvis) to obtain hierarchical clustering heatmaps. For heatmaps production, the parameters were set as: data import, upload file, detect delimiter, detect column and row annotations, no quotes, NA; pre-processing options: no transformation, maximum percentage for rows and columns, row centering, no scaling, Nipal PCA, PCA (no need), heatmap (generated accordingly). The graphs and tables were designed using Excel. The $IC_{50}$ and $BIC_{50}$ values were calculated from the respective percent inhibition values by using GraphPad Prism 5.0 software (San Diego, CA, USA). To determine $IC_{50}$, a log (inhibitor concentration) versus response non-linear fit was used. Post-hoc analysis using Tukey's Honestly Significant Difference (HSD) test was employed to determine the differences between the plant extracts prepared in different solvent groups and their antimicrobial activities. The Chi-square test was used to determine the association between the solvents and the antibiofilm activity (number of effective extracts) against *S. aureus* (USA 300) and *C. albicans* (SC 5314) as well as the association between solvents and cytotoxic activity (number of cytotoxic extracts) against A549 and WI 26 VA4 cell lines. A p-value of less than 0.05 was considered statistically significant.

# Results and discussion

The effects of traditional medicinal plants of Swat Pakistan against infectious diseases have gained more charm from phytomedicine researchers due to their effective use by the local community. Ethnobotanical researchers have documented more than 345 medicinal plants of Swat so far [12] but the bioactivities tests are limited. This work is an effort to enumerate the ethnobotanical data based on which extensively used and high-rated effective medicinal plants can be screened for the treatment of diseases.

## Documentation of medicinal plants

The ethnobotanical survey of local medicinal plants of the Swat region of Pakistan ("S1 File") was conducted, in which 17 plants shown in "Fig 1", belonging to 12 families were selected and collected from different locations of the study area. "Table 1" provides detailed information on these plants with botanical names, voucher number, local name, family, used plant part, GPS coordinates, and medicinal uses. Among the collected plants, 3 plants (*Ajuga bracteosa, Origanum vulgare,* and *Thymus linearis*) belonging to the Labiatae family, 3 plants (*Mentha longifolia, Salvia moorcroftiana,* and *Mentha spicata*) to Lamiaceae family, and 11 plants *viz. Debregeasia salicifolia*, *Berberis lyceum, Aesculus indica, Calotropis procera, Plantago major, Dysphania ambrosioides, Ziziphus oxyphylla, Punica granatum, Juglans regia, Artemisia maritima,* and *Nasturtium officinale* belonging to Urticaceae, Berberidaceae, Hippocastanaceae, Asclepiadaceae, Plantaginaceae,

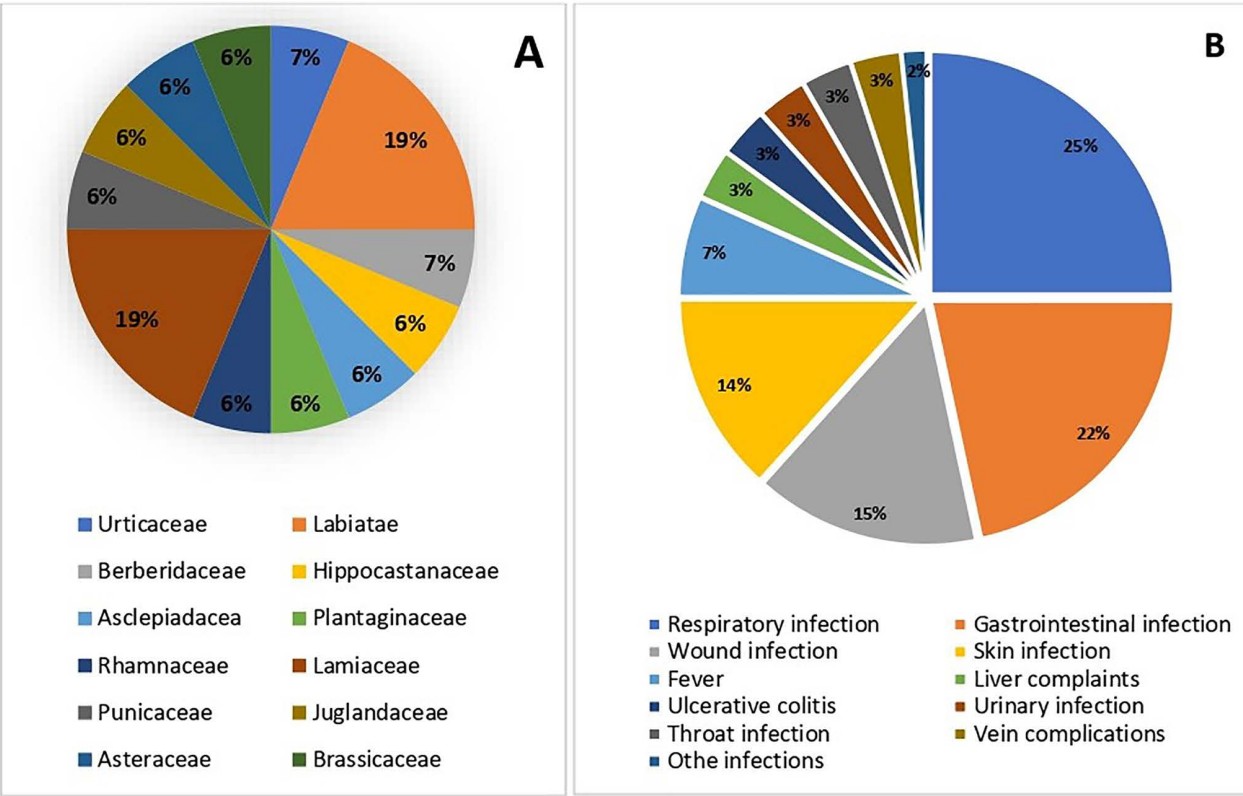

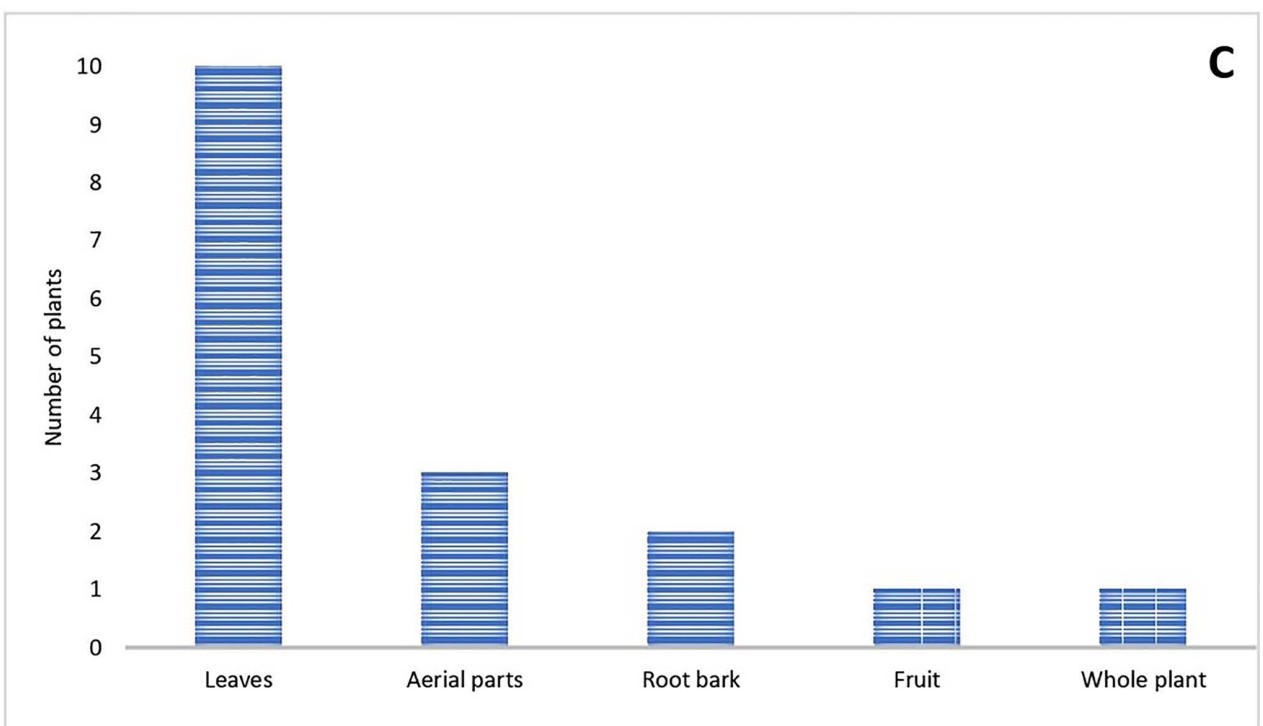

**Fig 2. Ethnomedicinal information.** (A) Distribution of the collected medicinal plants by taxonomic family; (B) Frequency of the traditional use of the collected plants in treating various infections; (C) Frequency of the collected plant parts.

Table 2. Phytochemical characterization of the tested medicinal plants.

| S. No. | Plant name | Alkaloids | Flavonoids | Phenols | Steroids | Terpenoids | Coumarins | Tannins | Saponins | Chalcones | Quinones |
|---|---|---|---|---|---|---|---|---|---|---|---|
| 1 | *Debregeasia salicifolia* | +++ | ++ | + | _ | ++ | + | + | _ | + | ++ |
| 2 | *Ajuga bracteosa* | +++ | +++ | ++ | + | +++ | + | ++ | + | _ | ++ |
| 3 | *Berberis lyceum* | ++ | +++ | +++ | +++ | + | +++ | + | + | ++ | +++ |
| 4 | *Aesculus indica* | ++ | ++ | + | _ | +++ | _ | +++ | ++ | + | + |
| 5 | *Calotropis procera* | ++ | + | + | +++ | + | + | + | +++ | _ | ++ |
| 6 | *Plantago major* | ++ | ++ | ++ | +++ | ++ | ++ | +++ | + | + | + |
| 7 | *Origanum vulgare* | ++ | + | ++ | + | +++ | + | ++ | + | + | + |
| 8 | *Dysphania ambrosioides* | ++ | +++ | + | + | + | + | ++ | + | _ | + |
| 9 | *Ziziphus oxyphylla* | ++ | ++ | ++ | ++ | +++ | + | ++ | _ | _ | + |
| 10 | *Thymus linearis* | +++ | +++ | +++ | ++ | ++ | ++ | ++ | ++ | + | + |
| 11 | *Mentha longifolia* | ++ | + | _ | + | ++ | + | + | + | _ | + |
| 12 | *Punica granatum* | ++ | + | ++ | + | + | +++ | + | +++ | ++ | +++ |
| 13 | *Juglans regia* | +++ | ++ | + | +++ | +++ | + | + | + | +++ | ++ |
| 14 | *Salvia moorcroftiana* | ++ | _ | + | + | + | + | + | + | _ | ++ |
| 15 | *Artemisia maritima* | + | ++ | + | ++ | ++ | + | ++ | ++ | _ | + |
| 16 | *Mentha spicata* | +++ | + | ++ | +++ | +++ | + | +++ | + | + | + |
| 17 | *Nasturtium officinale* | + | + | + | +++ | + | + | + | + | _ | + |

Legend. _: Absence; +: Low presence; ++: Moderate presence; +++: High presence.

Amaranthaceae, Rhamnaceae, Punicaceae, Juglandaceae, Asteraceae, and Brassicaceae family respectively as shown in "Fig 2A". The majority of the reported medicinal plants were found to be used by locals for the treatment of respiratory infections (15), gastrointestinal infections (13), wound infections (9), skin infections (8), fever (4), liver complaints (3), ulcerative colitis (2), urinary infections (2), throat infections (2), vein complications (2), or other infections like rheumatism, dental complaints, malaria, tuberculosis, headache, tumors, burns treatment, etc. ("Fig 2B"). Regarding the parts used, leaf was the mostly used part (10), followed by aerial parts (3), fruit, root bark, and whole plant (1 each) as evident from "Fig 2C".

Plants are considered as a rich bio-resource for several treatments in traditional and even modern medicine [14]. Globally, there are three major traditional medicines still practiced; Indian Ayurveda, traditional Chinese medicine (TCM), and Arabic medicine [26]. The medicinal uses of various plants are well-described in these major medicine systems, as well as in various European historical reports [14]. In Pakistan, India, and other parts of the world, tribal people are practicing traditional medicine for various ailments, and this indigenous knowledge needs to be preserved and validated through scientific research. Traditional medicines aid in providing precious clues for finding new drug candidates [27]. The leaves of *C. procera* are used for skin disorders and indigestions [28]. The leaves of *S. moorcroftiana* are used for cough, asthma, itching and guinea worm infections [11]. Nearly all the parts of *J. regia* like bark, kernel, flowers, leaves, green husk, septum, and oil have their medicinal applications [29]. The bark of *J. regia* is traditionally used for tooth cleaning, gum problems, toothaches, skin ailments, wounds, join pain and hair loss [30]. *A. maritima* is used for stomach ache, jaundice, whooping cough, and intermittent fever [31]. *B. lycium* roots are used for joint pain, rheumatism, chest infections, wounds, and fractured bones [32]. The leaves of *N. officinale* are traditionally used for the treatment of abdominal pain, influenza, asthma, and as anti-ulcerogenic [33]. The *D. salicifolia* is used traditionally for the treatment of several diseases like bone fractures, diarrhoea, dermatitis, skin rash, and eczema [34]. *Z. oxyphylla* is used for the treatment of fever, diabetes, and skin infections [35]. The leaves of *M. longifolia* are mostly used for coughs, colds, stomach cramps, asthma, flatulence, indigestion, and headaches [36]. The leaves of *O. vulgare* are used for asthma, cough, and cold [37]. The fruits

of *A. indica* are eaten raw for colics. The leaves of *A. bracteosa* are used for throat infections and fever. The fresh leaves of *P. major* are used for healing wounds, while the seeds are used in dysentery. The fruits of *P. granatum* are used for diarrhoea, dysentery, and whooping cough, and as laxatives and blood purifiers. The whole plant of *T. linearis* is used for stomach infections [38]. *M. spicata* is used for treating colds and flu, respiratory tract problems, gastralgia, hemorrhoids, and stomach ache [39]. In this study, we documented the traditional uses of the selected medicinal plants. However, due to limitations, we were not able to perform advanced analyses such as the molecular characterization of the plant extracts and in silico identification of active compounds. Some recent studies [40–43] have shown that *in silico* methods are very helpful for predicting biological activities and elucidating molecular mechanisms. Such studies will help better understand the medicinal properties of these plants and could lead to the development of novel bioactive agents. Therefore, it is strongly recommended that future studies should focus on detailed molecular studies and computational approaches for the identification and characterization of phytoconstituents. Ethnomedicinal literature on medicinal plants of Swat is available, but to the best of our knowledge as per available literature, the comprehensive bioactivities are lacking. Therefore, we documented comprehensively the *in vitro* antimicrobial activity of the most used and less studied plant extracts of the Swat region against planktonic as well as biofilm forms of bacteria and fungi. In addition to the antimicrobial activities, we also performed cytotoxicity assays on the crude extracts to select the most effective and non-cytotoxic plant extracts for further studies.

## Phytochemical screening

The phytochemical analysis of the ethanol extracts of the 17 traditional medicinal plants of the Swat region of Pakistan revealed the presence of various secondary metabolites, including alkaloids, flavonoids, phenols, steroids, terpenoids, coumarins, tannins, saponins, chalcones, and quinones as shown in "Table 2."

All tested extracts exhibited positive results for most of the tested secondary metabolites. The extracts of *B. lyceum*, *P. major*, *O. vulgare*, *T. linearis*, *P. granatum*, *J. regia*, and *M. spicata* contained all types of secondary metabolites. Similarly *A. bracteosa*, *C. procera*, *D. ambrosioides* also contained majority of the secondary metabolites. Alkaloids were detected in high concentration in majority of the extracts, while terpenoids, tannins and quinones were also detected in medium or low concentration in all plant extracts. Flavonoids were found in all plants except *S. moorcroftiana*. The phenolic compounds were absent in *M. longifolia*, while coumarins were absent in *A. indica* only. Steroids were absent in *D. salicifolia*, and *A. indica,* while saponins were absent in *D. salicifolia*, and *Z. oxyphylla*. On the other hand, chalcones were absent in many plant extracts like *A. bracteosa*, *C. procera*, *D. ambrosioides*, *Z. oxyphylla*, *M. longifolia*, *S. moorcroftiana*, *A. maritima*, and *N. officinale*. The quantitative analysis of the proportional distribution of each phytochemical constituent across 17 traditional medicinal plants from Swat region of Pakistan is shown in "S2 Table in S1 File." The presence of a large number of the tested secondary metabolites in the medicinal plants of Swat suggest that these plants contain a diverse range of bioactive compounds which are responsible for their activities. The results of the antimicrobial, antibiofilm and phytochemical screening of the tested plant extracts support the potential of these plants for the development of more effective antimicrobial agents.

The phytochemical screening of the plant materials is performed for unfolding the active principles which are responsible for the bioactivities of the plants as it provides the base for targeted isolation of active compounds [18]. The biological activities of plants are mainly due to the presence of secondary metabolites such as alkaloids, flavonoids, coumarins, steroids, phenolic compounds, terpenoids, glycosides, etc., which are distributed in various parts of the plants [19]. The secondary metabolites enable the plants to protect themselves against biotic and abiotic stresses, but have turned secondarily into medicines which are used to combat against pathogenic microorganisms and are responsible for the treatment of various infections [44]. The antipathogenic properties of the plants are dependent upon the concentration and interactions of the secondary metabolites [45]. Alkaloids are the most important secondary metabolites which constitute most of the drugs [46]. Flavonoids possess numerous health promoting effects and anti-pathogenic properties

[19]. Phenolic compounds have the capability to reduce inflammations [47]. Saponins are reported to as immune system booster in humans and protect the body against various microbes [48]. Steroids can serve as a potent starting materials in synthesis of sex hormones and thus help women in maintaining their hormonal balance during pregnancy or breast feeding [19]. Terpenoids act as promising phytochemicals by targeting cancer cells with high selectivity [49]. Tannins play an important role as defence compounds and also reduce the activity of many enzymes [50]. Coumarins possess anticancer, anti-inflammatory, anticoagulant, and anti-pharmacological Alzheimer's effects [51]. Quinones are important group of secondary metabolites that possess physiological and therapeutical effects mainly due to electron reduction or nucleophilic attack [52]. Chalcones possess a diverse range of biological effects like antimicrobial, antiviral, antioxidative, anthelmintic, antiparasitic, anticancer, and immunosuppressive [53].

Literature reports indicate that *A. bracteosa* contains kaempferol, rutin, quercetin, bractin A, bractin B, flavonol glycosides, iridoid glycosides, bractic acid, stigmasterol, ß-sitosterol, clerodin, ajugarin I, lupulin A, resorcinol, pyrocatechol, catechin, chlorogenic acid, gallic acid, caffeic acid, p-coumaric acid, syringic acid, ferulic acid, vanillic acid and coumarin, among which many of the compounds are known for antibacterial, antifungal, antiviral, anti-inflammatory, antidiabetic, antioxidant, anti-cancer, insecticidal, and cytotoxic activities [54–56]. Similarly, *O. vulgare* is rich in carvacrol, thymol, luteolin-O-glucuronide, luteolin-7-O-glucoside, caffeic acid, protocatechuic acid, vanillic acid, sabinene, γ-terpinene, linalool, borneol, and sesquiterpenes, with established antimicrobial, anti-inflammatory, antidiabetic, and antioxidant properties [57]. In *T. linearis*, compounds such as α-pinene, α-thujene, α-terpinene, p-cymene, γ-terpinene, camphene, myrcene, borneol, thymol, carvacrol, thymyl acetate, and β-bisabolene have been reported with known antibacterial and antifungal activities [58]. The plant *M. longifolia* is known for antimicrobial, anti-viral, antiparasitic, anti-inflammatory, anti-nociceptive, antioxidant, antipyretic, and insecticidal activities with reported compounds such as luteolin 7-O-glucoside, eriodicty ol-7-rutinoside; iso-orientin, rosmarinic acid, carvone, pulegone, menthol, sabinene, apigenin-7-O-glucoside, apigenin-7-O-rutinoside, apigenin-7-O-glucoronide, longitin, iso-orientin, lucenin-1, and vicenin-2 [59]. The anxiolytic properties of *S. moorcroftiana* are reportedly remarkable with a number of isolated compounds including sabinene, α-humulene, α-copaene, β-caryophyllene, (Z)-β-ocimene, germacrene D, and bicyclogermacrene [60]. Notably, *M. spicata* contains luteolin, rutin, limonene, catechin, epicatechin, pulegone, carvone, cis-carveol, trans-carveol, cis-dihydrocarvone, dihydrocarveol, β-bourbonene, myricetin, naringenin, and apigenin, with known reports of the antibiofilm, antidermatophytic, and pancreatic lipase inhibitory activity from majority of these compounds [61,62]. *D. salicifolia* has been testified to have compounds like lupeol, uvaol, stigmasterol, β-sitosterol, pomolic acid, tormentic acid, oleanolic acid, and ursolic acid, with discovered pharmacological activities including antibacterial, antifungal, anti-inflammatory and immune suppressant [34]. *B. lyceum* is rich in alkaloids and a number of other phytochemicals such as berberine, plamitine, berbamine, gilgitine, jhelumine, punjabine, sindamine, chinabine and umbellatine, which have been reported to have strong antimicrobial, anti-cancer, antiparasitic, antioxidant, and anti-inflammatory activities [63,64]. The plant *A. indica* contains aesin, esculin, rutin, quercetin, astragalin, enterolactone, methoxy chrysin, gallic acid, carnosic acid, p-coumaric acid, apigenic acid and ferulic acid, many of which have been reported to exhibit antibacterial, antifungal, anti-inflammatory, antitumor, antioxidant, anti-cancer activities [65–67]. *C. procera* contains various bioactive compounds such as kaempferol, steroidal, stigmasterol, catechin, calotropin, rutin, quercitrin, uscharin, calotoxin, calactin, proanthocyanidin, chlorogenic, hesperidin, naringin, germanicyl, mannosamine, tridecane, digitoxin, digitoxigenin, calotoxin, procesterol and pentatriacontane, which have been reported to have antimicrobial, anti-inflammatory and anticancer activities [67–69]. *P. major* contains bioactive compounds such as indicain, plantagonin, aucubin, aucubigenin, oleanolic acid, and ursolic acid, which are known for their antimicrobial, anti-inflammatory, antioxidants, and antiallergic activities [70]. The major bioactive constituents of *D. ambrosioides* include α-terpinene, piperitone, thymol, carvacrol, patulin, limonene, chrysin, scopoletin, Squalene, phytol, o-cymol, dimethyl phthalate, and β-sitosterol, which exhibit antimicrobial, antiparasitic, anti-inflammatory and antioxidant activities [71,72]. *Z. oxyphylla* contains bioactive compounds such as stigmasterol, betulinic acid, 5-pentadecanoic acid, and p-coumaric acid, which possess strong antioxidant activities [73,74]. Similarly, *P. granatum* is rich in compounds such as

kaempferol, catechin, quercetin, rutin, luteolin, cyanidin, punicalin, genistein, linoleic acid, cinnamic acid, and pelletierine, many of which are known for their antimicrobial, anti-inflammatory and antioxidant properties [75]. The major chemical constituents of *J. regia* include juglanin, myricetin, quercetin, kaempferol, isorhamnetin, procyanidin A, aloeresin G, and cassiasid, which possess antibacterial, antifungal, antioxidant, anti-inflammatory, anticancer, and antidiabetic properties [76]. *A. maritima* contains bioactive compounds such as artemin, vulgarin, maritimin, camphor, morin, ellagic acid, rutin, and pyrogallol, among which many have been reported to exhibit strong antioxidant activities [31,77]. *N. officinale* is rich in polyphenols and flavonoids such as kaempferol-3 (feroyl-triglucoside) 7-rhamnosyl, quercetin 3- (para coumaroyltruglucoside) 7- rhamnosyl, quercetin 3-triglucoside-7 rhamnoside, and kaempferol 3-triglucosie-7 rhamnoside, which contribute to its antioxidant, anti-inflammatory and anticancer activities [78]. These literature-reported compounds and their associated bioactivities support the traditional use of these plants, and provide a scientific basis for the pharmacological, antimicrobial and cytotoxic effects observed in this study.

## TLC profiling

The ethanol extracts of the selected plants were subjected to TLC plates for developing fingerprints which represent the phytochemicals like alkaloids, flavonoids, terpenoids, steroids, phenols, and other secondary metabolites. The number of the observed phytochemicals and their retention factor (Rf) values of the used medicinal plants are presented in "Table 3." The observed colors of different spots ("S2 File") and various Rf values indicate the presence of various types of secondary metabolites in the selected plants. These secondary metabolites or phytochemicals are responsible for the bioactivities of plants which were selected based on ethnomedicinal use and preliminary bioactivities, but the specific bioactive compound responsible for the activity needs to be studied further.

In this study, TLC was performed by using two different proportions of mobile phases, although, still some of the extracts showed similar fingerprints and Rf values, which suggest the presence of similar chemical classes. The Rf values also indicate that both polar and non-polar compounds are present in these plants. The observed spots and Rf values suggest the presence of various compounds in the selected plants and hence the activity shown by the crude extracts against various pathogens support further work for isolation and identification of the bioactive compounds. The separation of TLC is based on polarity and it is a convenient way of separating many compounds on a planar surface [22]. The Rf values above 0.5 indicate polar compounds, while the Rf values below 0.5 indicate the presence of non-polar compounds in the selected plants [79]. As compared to other chromatographic techniques, TLC is an easy way for exploring the characterization of the plant extracts in the form of fingerprints, which give a better direction of using advanced technique for isolating the specific bioactive compounds [21]. The TLC has another advantage of speeding up the identification of the active compounds from plant materials by exposing the test microorganisms to the TLC plates through contact bioautography [80].

## Antimicrobial effects

The antimicrobial activities of the 85 crude extracts from 17 traditional medicinal plants of Swat Pakistan were evaluated by microdilution broth assay. For the extraction of the used plant parts, 5 different solvents (hexane, acetone, ethanol, methanol, and water) were used. A total of 85 extracts were tested against 20 different human pathogens that include 4 Gram-positive bacteria, 10 Gram-negative bacteria, and 6 yeasts. All the experiments were performed twice, and the average percent inhibition values were calculated. The inhibition values of the antimicrobial activity of all 85 extracts against the pathogens were clustered and presented in a heatmap ("Fig 3"). Interestingly, 76 extracts showed pronounced activity (>50% growth inhibition) against at least one or a maximum of 19 microorganisms, accounting for 89% of the tested extracts as shown in the supporting information ("S1 File"). ("Tables S3–S5" in S1 File). In a total of 1700 tests, 426 (25.06%) showed pronounced activity, out of which the 409 most active ones were selected for determining their $IC_{50}$ ("Table 4"). Out of 409, 223 (54.5%)

**Table 3. Thin layer chromatographic (TLC) profiling of the tested medicinal plants.**

| S. No | Plant name | Distance travelled b | Distance 1 | Retention factor (Rf) |
|---|---|---|---|---|
| 1 | Debregeasia salicifolia | 1.8 | 6.5 | 0.2769 |
| | | 6.1 | 6.5 | 0.9385 |
| 2 | Ajuga bracteosa | 1.5 | 6.5 | 0.2308 |
| | | 2.7 | 6.5 | 0.4154 |
| | | 3.3 | 6.5 | 0.5077 |
| | | 5.7 | 6.5 | 0.8769 |
| 3 | Berberis lyceum | 0.3 | 6.5 | 0.0462 |
| | | 0.6 | 6.5 | 0.0923 |
| | | 1.5 | 6.5 | 0.2308 |
| | | 2.7 | 6.5 | 0.4154 |
| | | 3 | 6.5 | 0.4615 |
| 4 | Aesculus indica | 2.8 | 6.5 | 0.4308 |
| | | . | 4.7 | 6.5 | 0.7231 |
| | | 6 | 6.5 | 0.9231 |
| 5 | Calotropis procera | 2 | 6.5 | 0.3077 |
| | | 4.8 | 6.5 | 0.7385 |
| | | 5 | 6.5 | 0.7692 |
| 6 | Plantago major | 1.5 | 6.5 | 0.2308 |
| | | 2 | 6.5 | 0.3077 |
| | | 4.6 | 6.5 | 0.7077 |
| | | 6 | 6.5 | 0.9231 |
| 7 | Origanum vulgare | 2.3 | 6.5 | 0.3538 |
| | | 4.5 | 6.5 | 0.6923 |
| 8 | Dysphania ambrosioides | 4.6 | 6.5 | 0.7077 |
| | | 5.8 | 6.5 | 0.8923 |
| 9 | Ziziphus oxyphylla | 2.2 | 6.5 | 0.3385 |
| | | 4.7 | 6.5 | 0.7231 |
| 10 | Thymus linearis | 2.6 | 6.5 | 0.4 |
| | | 3.3 | 6.5 | 0.5077 |
| | | 5.4 | 6.5 | 0.8308 |
| | | 5.9 | 6.5 | 0.9077 |
| 11 | Mentha longifolia | 2.6 | 6.5 | 0.4 |
| | | 4.2 | 6.5 | 0.6462 |
| | | 4.7 | 6.5 | 0.7231 |
| | | 5.4 | 6.5 | 0.8308 |
| | | 5.9 | 6.5 | 0.9077 |
| 12 | Punica granatum | 2 | 6.5 | 0.3077 |
| | | 2.7 | 6.5 | 0.4154 |
| | | 5.8 | 6.5 | 0.8923 |
| 13 | Juglans regia | 2.4 | 6.5 | 0.3692 |
| | | 2.8 | 6.5 | 0.4308 |
| | | 3.1 | 6.5 | 0.4769 |
| | | 3.4 | 6.5 | 0.5231 |
| | | 6.2 | 6.5 | 0.9538 |
| 14 | Salvia moorcroftiana | 2.3 | 6.5 | 0.3538 |
| | | 2.7 | 6.5 | 0.4154 |

*(Continued)*

**Table 3.** (Continued)

| S. No | Plant name | Distance travelled b | Distance 1 | Retention factor (Rf) |
|---|---|---|---|---|
|  |  | 3.8 | 6.5 | 0.5846 |
|  |  | 4.5 | 6.5 | 0.6923 |
|  |  | 6.1 | 6.5 | 0.9385 |
| 15 | *Artemisia maritima* | 2.2 | 6.5 | 0.3385 |
|  |  | 2.8 | 6.5 | 0.4308 |
|  |  | 3.8 | 6.5 | 0.5846 |
|  |  | 6 | 6.5 | 0.9231 |
| 16 | *Mentha spicata* | 1.8 | 6.5 | 0.2769 |
|  |  | 2.2 | 6.5 | 0.3385 |
|  |  | 4.6 | 6.5 | 0.7077 |
|  |  | 6.1 | 6.5 | 0.9385 |
| 17 | *Nasturtium officinale* | 1.7 | 6.5 | 0.2615 |
|  |  | 2 | 6.5 | 0.3077 |
|  |  | 2.4 | 6.5 | 0.3692 |
|  |  | 4.6 | 6.5 | 0.7077 |

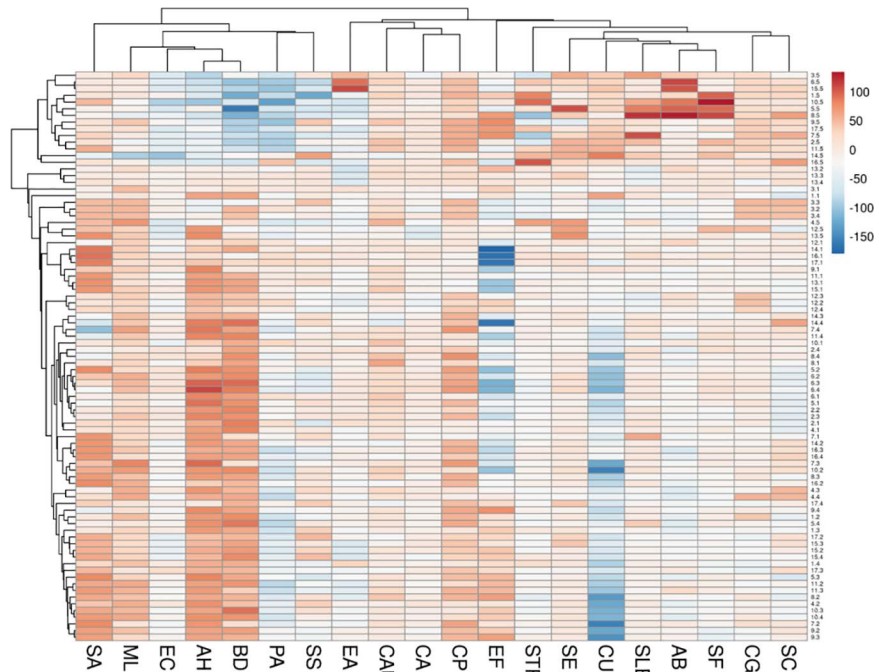

**Fig 3. Heat map and clustering of the inhibition values (IV) of antimicrobial activities of crude extracts against 20 microbes.** Legend. The vertical axis are 85 extracts of the 17 plants listed in Table 1 prepared in 5 solvents (.1−.5: hexane, acetone, ethanol, methanol, water). The horizontal axis are the abbreviations of 20 human pathogens. *Staphylococcus aureus* (SA); *Micrococcus luteus* (ML); *Escherichia coli* (EC); *Aeromonas hydrophila* (AH); *Brevundimonas diminuta* (BD); *Pseudomonas aeruginosa* (PA); *Shigella sonnei* (SS); *Enterobacter aerogenes* (EA); *Candida auris* (CAU); *Candida albicans* (CA); *Candida parapsilosis* (CP); *Enterococcus faecalis* (EF); *Streptococcus faecalis* (STF); *Staphylococcus epidermidis* (SE); *Candida utilis* (CU); *Salmonella enteritidis* (SLE); *Acinetobacter baumannii* (AB); *Shigella flexneri* (SF); *Candida glabrata* (CG); and *Saccharomyces cerevisiae* (SC).

**Table 4. IC$_{50}$ values (µg/mL) of the antimicrobial activity of the most active plant extracts against 20 different microorganisms.**

| Herb No. | Plant name | Gram-positive bacteria | | | | | Gram-negative bacteria | | | | | | | | | | Fungi | | | | |
|---|---|---|---|---|---|---|---|---|---|---|---|---|---|---|---|---|---|---|---|---|---|
| | | SA | SE | ML | STF | EF | EC | BD | SLE | EA | AB | SF | SS | AH | PA | CA | CAU | CG | CU | CP | SC |
| 1.1 | *Debregeasia salicifolia* | _ | _ | _ | _ | _ | _ | **137** | _ | _ | _ | _ | _ | 597 | _ | _ | _ | _ | **8** | _ | _ |
| 1.2 | | _ | _ | 594 | _ | _ | _ | **59** | _ | _ | _ | _ | _ | 666 | _ | _ | _ | 238 | _ | **146** | _ |
| 1.3 | | _ | _ | 555 | _ | **12** | _ | **37** | _ | _ | _ | _ | 461 | 470 | _ | _ | _ | _ | _ | _ | _ |
| 1.4 | | 496 | _ | **77** | 86 | 40 | _ | **60** | _ | 400 | _ | _ | _ | 474 | _ | _ | _ | _ | _ | _ | _ |
| 2.1 | *Ajuga bracteosa* | _ | _ | _ | _ | _ | _ | 334 | _ | _ | _ | _ | _ | 821 | _ | _ | _ | _ | _ | _ | _ |
| 2.2 | | _ | _ | _ | _ | _ | _ | 261 | _ | _ | _ | _ | _ | 689 | _ | _ | _ | _ | _ | _ | _ |
| 2.3 | | _ | _ | 952 | _ | _ | _ | 500 | _ | _ | _ | _ | _ | 625 | _ | _ | _ | _ | _ | 212 | _ |
| 3.2 | *Berberis lyceum* | 409 | _ | **73** | _ | _ | _ | _ | _ | _ | _ | _ | _ | _ | 793 | _ | 264 | **175** | _ | **156** | **188** |
| 3.3 | | 345 | _ | 217 | _ | _ | _ | _ | _ | _ | _ | _ | 480 | _ | _ | _ | 222 | **115** | _ | **182** | 203 |
| 3.4 | | 389 | _ | **142** | _ | _ | _ | 235 | _ | _ | _ | _ | _ | _ | _ | 376 | 277 | **88** | _ | 216 | **74** |
| 3.5 | | _ | 197 | _ | _ | _ | _ | _ | **124** | _ | _ | _ | _ | _ | _ | _ | _ | _ | _ | _ | _ |
| 4.1 | *Aesculus indica* | _ | _ | 422 | _ | _ | _ | **85** | _ | _ | _ | _ | _ | 467 | _ | _ | _ | _ | _ | _ | _ |
| 4.2 | | 828 | _ | 256 | _ | _ | _ | **8** | _ | _ | _ | _ | _ | 578 | _ | _ | _ | _ | _ | _ | _ |
| 4.3 | | _ | _ | 219 | _ | _ | _ | **36** | _ | _ | _ | _ | 648 | 515 | _ | _ | 338 | _ | _ | _ | **139** |
| 4.4 | | 488 | _ | **163** | _ | _ | _ | **97** | _ | _ | _ | _ | _ | 325 | _ | _ | 337 | **197** | _ | _ | **102** |
| 4.5 | | 374 | **143** | 90 | 41 | _ | _ | _ | _ | _ | _ | _ | _ | _ | _ | _ | 79 | _ | _ | _ | _ |
| 5.1 | *Calotropis procera* | _ | _ | _ | _ | _ | _ | **88** | _ | _ | _ | _ | _ | 843 | _ | _ | _ | _ | _ | **195** | _ |
| 5.2 | | 287 | _ | _ | _ | _ | _ | **163** | _ | _ | _ | _ | _ | 585 | _ | _ | _ | _ | _ | 267 | _ |
| 5.3 | | **121** | _ | 1086 | _ | **65** | _ | 352 | _ | _ | _ | _ | _ | 593 | _ | _ | _ | _ | _ | 238 | _ |
| 5.4 | | _ | _ | _ | _ | _ | _ | 319 | 518 | _ | _ | _ | _ | 660 | _ | _ | _ | _ | _ | **196** | _ |
| 5.5 | | _ | **103** | _ | _ | _ | _ | _ | 251 | _ | **70** | 206 | _ | _ | _ | _ | _ | _ | _ | _ | _ |
| 6.1 | *Plantago major* | | _ | _ | _ | _ | _ | 500 | _ | _ | _ | _ | _ | 304 | _ | _ | 349 | _ | _ | _ | _ |
| 6.2 | | 342 | _ | 673 | _ | _ | _ | 267 | _ | _ | _ | _ | _ | 613 | _ | _ | _ | _ | _ | 285 | _ |
| 6.3 | | _ | _ | _ | _ | _ | _ | 252 | _ | _ | _ | _ | _ | 651 | _ | _ | _ | _ | _ | 205 | _ |
| 6.4 | | _ | _ | _ | _ | _ | _ | 453 | _ | _ | _ | _ | _ | 526 | _ | _ | _ | _ | _ | 282 | _ |
| 6.5 | | _ | _ | _ | _ | _ | _ | _ | _ | 353 | **64** | _ | _ | _ | _ | _ | _ | _ | _ | _ | _ |
| 7.1 | *Origanum vulgare* | 256 | _ | 984 | _ | _ | _ | _ | **468** | _ | _ | _ | 767 | 463 | _ | _ | _ | _ | _ | _ | _ |
| 7.2 | | 240 | _ | _ | _ | _ | _ | _ | _ | _ | _ | _ | _ | 637 | _ | _ | _ | _ | _ | **75** | _ |
| 7.3 | | 572 | _ | 329 | _ | _ | _ | _ | _ | _ | _ | _ | _ | 532 | _ | _ | _ | _ | _ | **152** | _ |
| 7.4 | | _ | _ | 321 | _ | _ | _ | 498 | _ | _ | _ | _ | _ | 543 | _ | _ | _ | _ | _ | 220 | _ |
| 7.5 | | _ | _ | _ | _ | _ | _ | _ | **161** | _ | _ | _ | _ | _ | _ | _ | _ | _ | _ | _ | _ |
| 8.1 | *Dysphania ambrosioides* | _ | _ | **20** | _ | _ | _ | 495 | _ | _ | _ | _ | _ | 1370 | _ | _ | **155** | _ | _ | 301 | _ |
| 8.2 | | 241 | _ | _ | _ | **57** | _ | 300 | _ | _ | _ | _ | 871 | 535 | _ | _ | **146** | _ | _ | 233 | _ |
| 8.3 | | **145** | _ | 412 | _ | _ | _ | 304 | _ | _ | _ | _ | _ | 627 | _ | _ | 302 | _ | _ | **97** | _ |
| 8.4 | | _ | _ | _ | _ | _ | _ | 212 | _ | _ | _ | _ | _ | _ | _ | _ | 383 | _ | _ | **114** | _ |
| 8.5 | | _ | _ | _ | _ | _ | _ | _ | **51** | _ | **51** | 156 | _ | _ | _ | _ | _ | _ | _ | _ | _ |
| 9.1 | *Ziziphus oxyphylla* | **99** | _ | 1422 | _ | _ | _ | _ | _ | _ | _ | _ | _ | 522 | _ | _ | _ | _ | _ | _ | _ |
| 9.2 | | **126** | _ | 514 | _ | **26** | _ | _ | _ | _ | _ | _ | _ | 599 | _ | _ | _ | _ | _ | **198** | _ |
| 9.3 | | 304 | _ | 915 | _ | **5** | _ | 313 | _ | _ | _ | _ | _ | 602 | _ | _ | _ | _ | _ | **160** | _ |
| 9.4 | | _ | _ | _ | _ | **20** | _ | 329 | _ | _ | _ | _ | _ | 629 | _ | _ | _ | _ | _ | **131** | _ |
| 10.1 | *Thymus linearis* | | _ | 758 | _ | _ | _ | 388 | _ | _ | _ | _ | _ | 1043 | _ | _ | 240 | _ | _ | _ | _ |

*(Continued)*

**Table 4.** (Continued)

| Herb No. | Plant name | Gram-positive bacteria | | | | | Gram-negative bacteria | | | | | | | | | | Fungi | | | | |
|---|---|---|---|---|---|---|---|---|---|---|---|---|---|---|---|---|---|---|---|---|---|
| | | SA | SE | ML | STF | EF | EC | BD | SLE | EA | AB | SF | SS | AH | PA | CA | CAU | CG | CU | CP | SC |
| 10.2 | | 225 | _ | **167** | _ | _ | _ | **73** | _ | _ | _ | _ | _ | 628 | _ | _ | **184** | _ | _ | _ | _ |
| 10.3 | | 874 | _ | 291 | _ | 221 | _ | 295 | _ | _ | _ | _ | 816 | 599 | _ | _ | _ | _ | _ | _ | _ |
| 10.4 | | 281 | _ | 420 | _ | 429 | _ | **81** | _ | _ | _ | _ | _ | 330 | _ | _ | _ | _ | _ | _ | _ |
| 10.5 | | _ | _ | _ | 39 | _ | _ | _ | 459 | _ | **18** | 37 | _ | _ | _ | _ | _ | _ | _ | _ | _ |
| 11.1 | *Mentha longifolia* | 382 | _ | _ | _ | _ | _ | 255 | _ | _ | _ | _ | _ | 576 | _ | _ | _ | _ | _ | _ | _ |
| 11.2 | | 274 | _ | **194** | _ | 92 | _ | 169 | _ | _ | _ | _ | _ | 631 | _ | _ | _ | _ | _ | 187 | _ |
| 11.3 | | 498 | _ | 320 | _ | 207 | _ | 227 | _ | _ | 914 | _ | _ | 591 | _ | _ | **152** | _ | _ | 250 | _ |
| 11.4 | | _ | _ | _ | _ | _ | _ | 499 | _ | _ | _ | _ | _ | 797 | _ | _ | _ | _ | _ | _ | _ |
| 12.1 | *Punica granatum* | | _ | 866 | _ | _ | _ | 395 | _ | _ | 485 | _ | _ | _ | _ | _ | _ | _ | _ | _ | _ |
| 12.2 | | **40** | _ | **111** | _ | 133 | _ | **3** | _ | _ | 273 | _ | _ | 460 | 697 | _ | **52** | 29 | _ | 135 | _ |
| 12.3 | | **44** | _ | 69 | 18 | _ | _ | 20 | _ | _ | **179** | _ | _ | 356 | 508 | _ | _ | 35 | _ | 147 | _ |
| 12.4 | | **13** | 131 | 84 | 31 | 110 | _ | 35 | _ | _ | 389 | _ | _ | 313 | 738 | _ | 110 | 32 | _ | 229 | _ |
| 12.5 | | 349 | **116** | 431 | _ | _ | _ | _ | _ | _ | _ | 246 | _ | _ | 256 | _ | _ | 70 | _ | _ | _ |
| 13.1 | *Juglans regia* | 291 | _ | _ | _ | _ | _ | **167** | _ | _ | _ | _ | _ | 273 | _ | _ | _ | _ | _ | _ | _ |
| 13.2 | | **39** | 284 | 5 | 24 | 46 | 485 | **14** | _ | _ | 100 | 85 | 149 | 63 | 448 | _ | 47 | 114 | 2 | 106 | 35 |
| 13.3 | | **34** | 130 | 1 | 14 | 79 | 235 | **15** | 298 | _ | 99 | 92 | 87 | 74 | 491 | 64 | 30 | 98 | 3 | 120 | 50 |
| 13.4 | | **43** | 144 | _ | 21 | 47 | 268 | **58** | 759 | _ | 95 | 120 | 105 | 57 | 403 | 136 | 24 | 141 | 7 | 142 | 102 |
| 13.5 | | **60** | 52 | 503 | _ | _ | _ | _ | _ | _ | _ | _ | _ | _ | 100 | _ | _ | _ | _ | _ | _ |
| 14.1 | *Salvia moorcroftiana* | 385 | _ | _ | _ | _ | _ | 290 | _ | _ | _ | _ | _ | _ | _ | 107 | _ | _ | _ | _ | _ |
| 14.2 | | 498 | _ | 714 | _ | _ | _ | **1** | _ | _ | _ | _ | _ | 622 | _ | _ | _ | _ | _ | 208 | 360 |
| 14.3 | | _ | _ | 434 | _ | _ | _ | 449 | _ | _ | _ | _ | _ | 493 | _ | _ | _ | _ | _ | 224 | _ |
| 14.4 | | _ | _ | _ | _ | _ | _ | 464 | _ | _ | _ | _ | _ | 319 | _ | _ | _ | _ | _ | _ | 238 |
| 14.5 | | _ | 207 | _ | 36 | _ | _ | _ | 193 | _ | _ | 137 | 92 | _ | _ | _ | _ | _ | 16 | _ | _ |
| 15.1 | *Artemisia maritima* | 288 | _ | 654 | _ | _ | _ | **60** | _ | _ | _ | _ | _ | 590 | _ | _ | 213 | _ | _ | _ | _ |
| 15.2 | | **130** | _ | 484 | _ | 110 | _ | **95** | _ | _ | _ | _ | 569 | 597 | _ | _ | **198** | _ | _ | 242 | _ |
| 15.3 | | 221 | 183 | 568 | _ | 845 | _ | **136** | _ | _ | _ | _ | 600 | 620 | _ | _ | _ | _ | _ | **160** | 290 |
| 15.4 | | 298 | _ | 900 | _ | **140** | _ | **84** | _ | _ | _ | _ | 227 | 698 | _ | _ | _ | _ | _ | _ | _ |
| 15.5 | | _ | _ | _ | 70 | _ | _ | _ | _ | 229 | **139** | _ | _ | _ | _ | _ | _ | _ | _ | _ | _ |
| 16.2 | *Mentha spicata* | **172** | _ | 494 | _ | _ | _ | **117** | _ | _ | _ | _ | _ | 941 | _ | _ | _ | _ | _ | 221 | _ |
| 16.3 | | **148** | _ | 410 | _ | _ | _ | **111** | _ | _ | _ | _ | _ | 767 | _ | _ | _ | _ | _ | 221 | _ |
| 16.4 | | 395 | _ | _ | _ | _ | _ | **159** | _ | _ | _ | _ | _ | 639 | _ | _ | _ | _ | _ | 341 | 290 |
| 17.1 | *Nasturtium officinale* | 258 | _ | _ | _ | _ | _ | _ | _ | _ | _ | _ | _ | 1251 | _ | _ | _ | _ | _ | _ | _ |
| 17.2 | | **111** | 149 | 468 | _ | **147** | _ | **61** | _ | _ | _ | _ | 294 | 564 | _ | _ | _ | _ | _ | **193** | 266 |
| 17.3 | | **94** | _ | _ | _ | 212 | 834 | **197** | _ | _ | _ | _ | _ | 569 | _ | _ | _ | _ | _ | 243 | _ |
| 17.4 | | 295 | 319 | _ | _ | 552 | _ | **86** | _ | _ | _ | _ | 632 | _ | _ | _ | _ | _ | _ | **158** | 273 |
| Positive control | | 0.28 | 0.49 | 2.11 | _ | 9.44 | 0.02 | 2.26 | 0.01 | 0.04 | 0.17 | 0.02 | 0.02 | 0.01 | 0.02 | 0.01 | 0.10 | 0.12 | _ | 0.13 | 0.01 |

Legend. No. 1–17: Plants mentioned; sub-number (−1 to −5): hexane, acetone, ethanol, methanol, water; SA: *Staphylococcus aureus*; SE: *Staphylococcus epidermidis*; ML: *Micrococcus luteus*; STF: *Streptococcus faecalis*; EF: *Enterococcus faecalis*; EC: *Escherichia coli*; BD: *Brevundimonas diminuta*; SLE: *Salmonella enteritidis*; EA: *Enterobacter aerogenes*; AB: *Acinetobacter baumannii*; SF: *Shigella flexneri*; SS: *Shigella sonnei*; AH: *Aeromonas hydrophila*; PA: *Pseudomonas aeruginosa*; CA: *Candida albicans*; CAU: *Candida auris*; CG: *Candida glabrata*; CU: *Candida utilis*; CP: *Candida parapsilosis*; and SC: *Saccharomyces cerevisiae* (SC).

showed IC$_{50}$ values between 200 µg/mL to 1000 µg/ mL, which is moderately active for crude extracts, while only 3 (0.73%) had values above 1000 µg/mL which is less active. Encouragingly, 183 test samples (44.7%) showed stronger inhibition with IC$_{50}$ values below 200 µg/mL, rendering the respective plants attractive for bioassay-guided purification and further studies, such as *J. regia*, *P. granatum*, *A. maritima*, *B. lyceum*, *A. indica*, *D. ambrosioides*, *T. linearis,* and *N. officinale.* The extracts of *J. regia* showed pronounced inhibitory activity (IV > 50%) against different microbes 63%, followed by *P. granatum* (41%), *A. maritima* (31%), *B. lycium* (27%), *A. indica* and *D. ambrosioides* (26%), *T. linearis* and *N. officinale* (25%), *D. salicifolia, C. procera,* and *S. moorcroftiana* (21%), *Z. oxyphylla* (20%), *M. longifolia* (19%), *P. major, O. vulgare,* and *M. spicata* (17%), while *A. bracteosa* showed pronounced against 9% of the microbes only ("Fig 4A"). The susceptibility of microorganisms to the plant extracts is presented in "Fig 4B". Out of 85 extracts, most activities were detected against *B. diminuta* and *A. hydrophila* (60 extracts), *M. luteus* (50), *S. aureus* (47), *C. parapsilosis* (42), *E. faecalis* and *C. auris* (22), *S. sonnei* (16), while the least susceptible microbes were detected to be *E. coli, E. aerogenes,* and *C. auris,* which were only inhibited by 4 extracts each. For *E. coli*, the acetone, ethanol, and methanol extracts of *J. regia* and methanol extracts of *N. officinale* showed prominent activity. Similarly, for *E. aerogenes*, methanol extracts of *D. salicifolia,* and water extracts of *B. lyceum, P. major,* and *A. maritima* showed pronounced activity; while for *C. auris* methanol extracts of *D. salicifolia,* and acetone, ethanol, and methanol extract of *J. regia* showed strong inhibition activity. The bioactivity data of all the extracts indicate a relationship between extracting solvents and bioactivities. Ethanol is the best overall solvent. As shown in "Fig 4C", the ethanol (116 extracts; 27.2%), acetone (115 extracts; 27%), and methanol (101 extracts; 24.7%) extracts are more active than hexane (52 extracts; 12.2%) or water extracts (42 extracts; 10%). The antimicrobial activity of plant extracts varied significantly depending on the solvent used (p = 0.003). The post-hoc analysis using Tukey's HSD test revealed significant differences between acetone and hexane (p < 0.001), acetone and water (p < 0.001), ethanol and hexane (p < 0.001), ethanol and water (p < 0.001), methanol and hexane (p = 0.002), and methanol and water (p < 0.001). Although, no significant differences were observed between acetone and ethanol (p = 0.001), acetone and methanol (p = 0.423), or ethanol and methanol (p = 0.423). Our results indicate that *J. regia*, *P. granatum*, *A. maritima*, *B. lyceum*, *A. indica*, *D. ambrosioides*, *T. linearis,* and *N. officinale* could be potential sources for broad-spectrum antibiotics as they inhibit the majority of the tested human pathogens like *S. epidermidis, B. diminuta, A. hydrophila, M. luteus, C. parapsilosis* and *S. aureus.* Moreover, plants such as *D. salicifolia, Z. oxyphylla, M. longifolia*, *C. procera, O. vulgare, S. moorcroftiana, M. spicata, P. major,* and *A. bracteosa* could be a source for narrow-spectrum antibiotics.

Based on review of the literature, that the detailed study on the antimicrobial activity was lacking. This gap is filled by the current study, and the comprehensive activities of all the selected medicinal plants of Swat have been documented in this report. After looking into various search engines, we came to know that so far the activity of *J. regia* [81–83], *P. granatum* [84–86], *A. maritima* [87–89], *B. lyceum* [90–92], *N. officinale* [93–95], *D. ambrosioides* [96–99], *D. salicifolia* [100–102], *A. indica* [102–104], *Z. oxyphylla* [35,105,106], *T. linearis* [58,107], *M. longifolia* [108,109], *C. procera* [110,111], *O. vulgare* [112,113], *S. moorcroftiana* [114,115], *M. spicata* [116–118], *P. major* [119,120], and *A. bracteosa* [121,122] has been reported in one or two solvents against some microbes like *S. aureus, S. epidermidis, E. coli, K. pneumoniae, P. aeruginosa,* and *C. albicans.* The current project has filled many gaps in the antimicrobial effects of the selected medicinal plants in five different solvents (from non-polar to polar) against 20 human pathogens. The reported potent antimicrobial and antibiofilm plant extracts can be a promising starting point for new antimicrobial drugs, including with antibiofilm activity. While determining the diverse mechanism of action(s) of bioactive compounds, the researchers are encouraged to explore structure activities relationship (SAR) and develop schematic classification based on their activities [123]. Further studies are recommended to identify and isolate the bioactive compounds from the promising plants with potential against the susceptible pathogens.

## Antibiofilm assessment

The antibiofilm effects of the plant extracts against *S. aureus* (USA 300) and *C. albicans* (SC 5314) are presented in the form of a heatmap in "Fig 5A" and "S1 File". Of the 85 tested plant extracts, 47 extracts of 16 plants showed antibiofilm activity

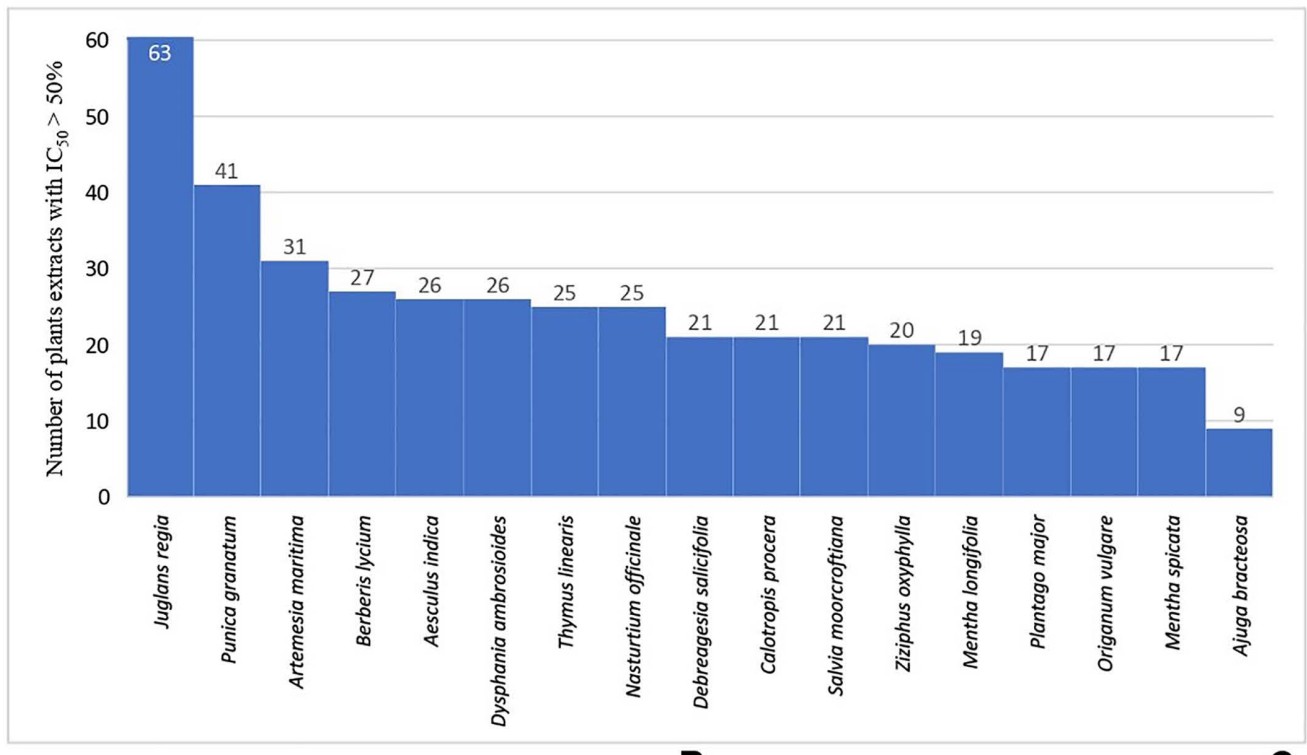

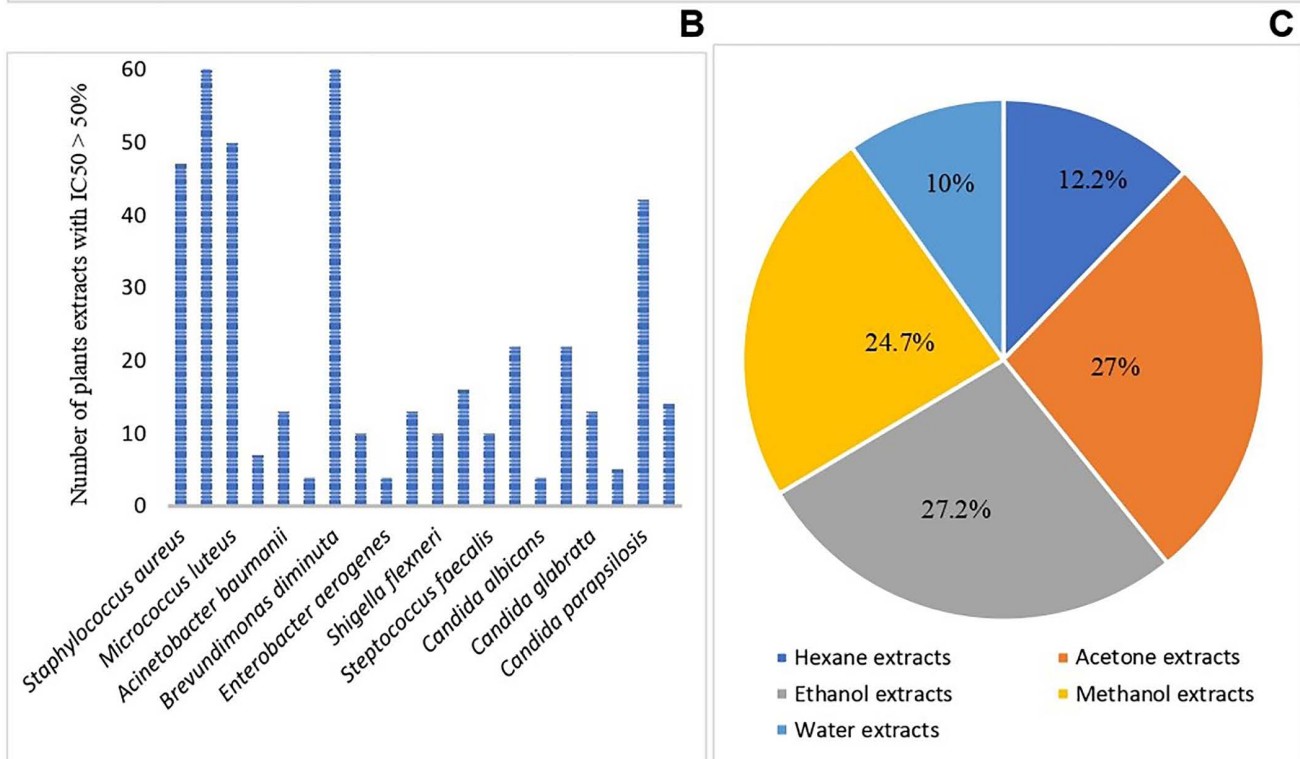

**Fig 4. Antimicrobial activities.** (A) Plant extracts with pronounced inhibitory activity against various microbes; (B) Susceptibility of microbes to the plant extracts with inhibition value ≥ 50%; (C) Pie chart showing percentage of active extracts by solvent used.

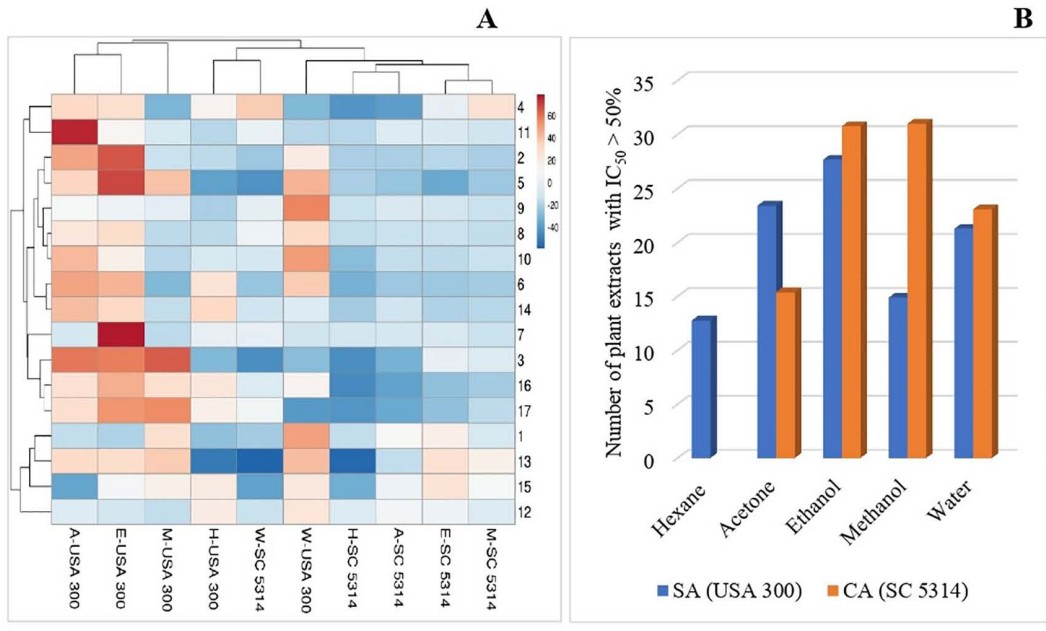

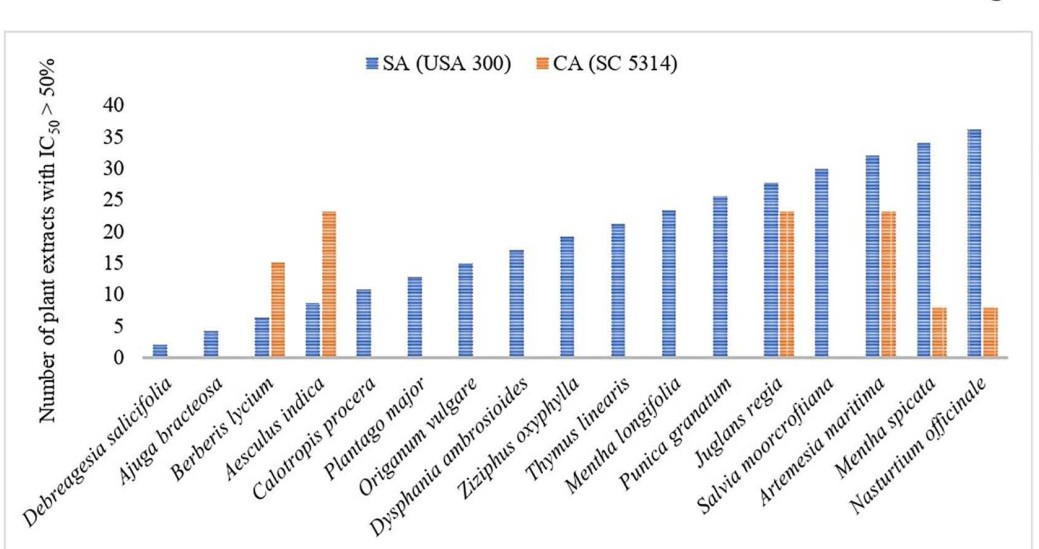

**Fig 5. Antibiofilm activities.** (A) Heat map and clustering of the inhibition values of antibiofilm effects of crude extracts against biofilm forming strains of *S. aureus* (USA 300) and *C. albicans* (SC 5314). Extracts were prepared in hexane (H), acetone (A), ethanol (E), methanol (M), or water (W); (B) Number of plant extracts with antibiofilm activity dependent on solvent; (C) Plant extracts with pronounced antibiofilm activity.

against *S. aureus* biofilm, while only 13 extracts of 5 plants effective against *C. albicans* biofilm strain. For *S. aureus* biofilm, all the five extracts of *M. spicata* (H,A,E,M,W); 4 extracts of each *J. regia* (A,E,M,W), *A. maritima* (H,E,M,W), *C. procera* (A,E,M,W) and *P. major* (H,A,E,W); 3 extracts of each *A. bracteosa* (A,E,W), *B. lyceum* (A,E,M), *A. indica* (H,A,E), *T. linearis* (A,E,W), *S. moorcroftiana* (H,A,E); 2 extracts of each *D. salicifolia* (M,W), and *D. ambrosioides* (E,W), while 1 extract of each *O. vulgare* (E), *Z. oxyphylla*, (W), and *M. longifolia* (A) were found active. Similarly, for *C. albicans*, 3 extract of *J. regia*

**Table 5. IC$_{50}$ values (µg/mL) of the antibiofilm activity of most active extracts against *S. aureus* (USA300) and *C. albicans* (SC 5314).**

| Herb No. | USA300 | | | | | SC 5314 | | | | |
|---|---|---|---|---|---|---|---|---|---|---|
| | H | A | E | M | W | H | A | E | M | W |
| 1 | _ | _ | _ | 465 | **192** | _ | _ | _ | _ | _ |
| 2 | _ | **114** | _ | _ | 667 | _ | _ | _ | _ | _ |
| 3 | _ | **29** | **34** | **54** | _ | _ | _ | _ | 199 | _ |
| 4 | 240 | **85** | 151 | _ | _ | _ | _ | 221 | 39 | 32 |
| 5 | _ | 305 | **43** | 171 | 91 | _ | _ | _ | _ | _ |
| 6 | **143** | 101 | 173 | _ | 263 | _ | _ | _ | _ | _ |
| 7 | _ | _ | 89 | _ | _ | _ | _ | _ | _ | _ |
| 8 | _ | 767 | 771 | _ | 477 | _ | _ | _ | _ | _ |
| 9 | _ | _ | _ | _ | 105 | _ | _ | _ | _ | _ |
| 10 | _ | 93 | 604 | _ | 197 | _ | _ | _ | _ | _ |
| 11 | _ | 77 | _ | _ | _ | _ | _ | _ | _ | _ |
| 12 | _ | _ | _ | _ | _ | _ | _ | _ | _ | _ |
| 13 | _ | 117 | 90 | 97 | 122 | _ | 289 | **26** | 38 | _ |
| 14 | 255 | 214 | 224 | _ | _ | _ | _ | _ | _ | _ |
| 15 | **162** | _ | 284 | **167** | 164 | _ | _ | 157 | 48 | 48 |
| 16 | 290 | 285 | **89** | 156 | 323 | _ | _ | _ | _ | 226 |
| 17 | 342 | 209 | **86** | 67 | _ | _ | _ | _ | 203 | _ |
| P* | 0,25 | _ | _ | _ | _ | 0,44 | _ | _ | _ | _ |

Legend. *Positive Control: Erythromycin for *S. aureus* (USA 300) and miconazole for *C. albicans* (SC 5314).

(A,E,M), *A. maritima* (A,E,M), *A. indica* (E,M,W); 2 extracts of *B. lyceum* (E,M); and 1 extract of each *M. spicata* (W) and *N. officinale* (W) were reported active. Overall, ethanol is the best solvent for biofilm strains like planktonic microbes, followed by methanol and acetone ("Fig 5B"). The Chi-square test results revealed that there is a significant association between the solvents used and the number of effective extracts against *S. aureus* (USA 300) with p-value = 0.004, and *C. albicans* (SC 5314) with p-value < 0.001. The effect of the plant extracts on both the biofilm strains of *S. aureus* and *C. albicans* is evident from "Fig 5C". The most active extracts were tested in two-fold serial dilutions in DMSO/water. The inhibition curves were analyzed by GraphPad Prism 5.0 Software and the BIC$_{50}$ values were determined. Interestingly, for *S. aureus* 13 extracts and for *C. albicans,* 6 extracts showed stronger inhibition with BIC$_{50}$ less than 100 µg/mL ("Table 5"). The positive controls were erythromycin (for *S. aureus;* USA 300) and miconazole (for *C. albicans;* SC 5314) with BIC$_{50}$ of 0.25 µg/mL and 0.44 µg/mL, respectively.

The antimicrobial activities of extracts against planktonic pathogens extended to biofilms for crude extracts of the plants; *J. regia* [81], *O. vulgare* [124], and *M. spicata* [81,116]. The effectiveness of these plant extracts against the biofilm forms also support their traditional use against various infections. Based on review of the literature, the detailed study on the antibiofilm activity was lacking. The antibiofilm activity of *J. regia*, *O. vulgare,* and *M. spicata* have been reported previously against biofilm strains of *S. aureus, S. epidermidis, E. coli, K. pneumoniae, P. aeruginosa,* and *C. albicans*. Although, to the best of our knowledge, the antibiofilm tests of plants like *A. maritima, B. lyceum, N. officinale, D. ambrosioides, D. salicifolia, A. indica, Z. oxyphylla, T. linearis, C. procera,* and *S. moorcroftiana,* have not been performed earlier by other researchers, and is reported here for the first time. The current project has filled many gaps in the antibiofilm effects of the selected medicinal plants in different solvents against *S. aureus* (USA 300) and *C. albicans* (SC 5314).

## Cytotoxicity activity

The *in vitro* cytotoxicity activity of all 85 plant extracts was investigated against a human lung epithelial cancerous cell line (A549) and noncancerous lung fibroblast cell line (WI-26 VA4) using a resazurin assay. The data were analyzed

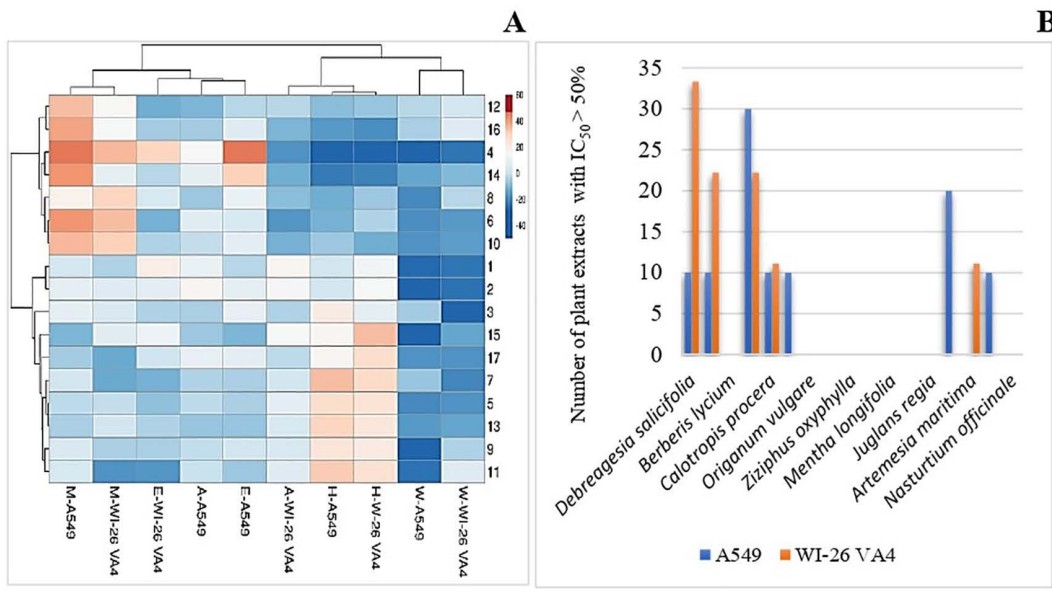

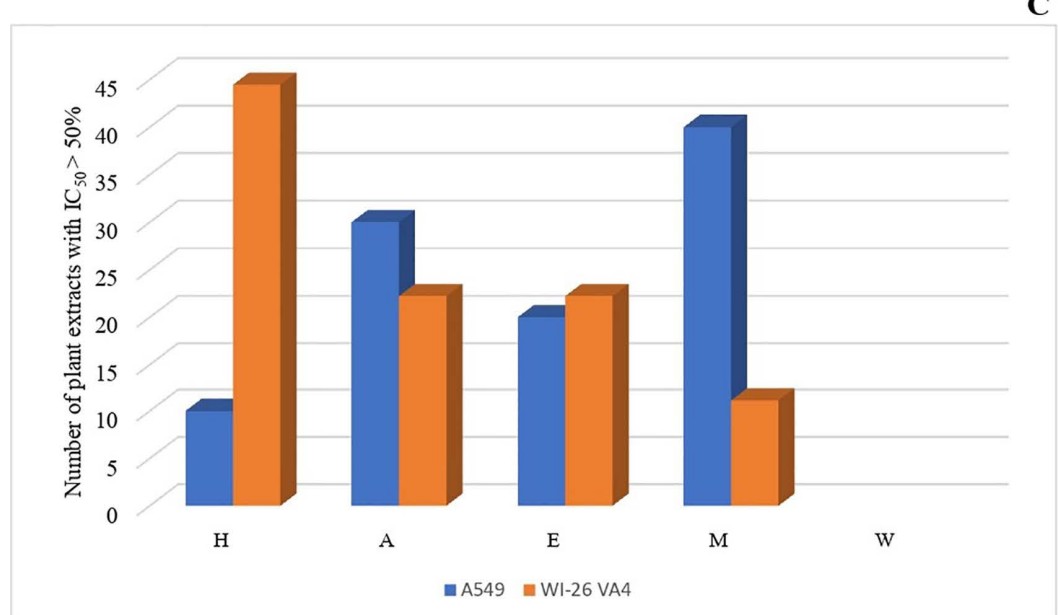

**Fig 6. Cytotoxicity activities.** (A) Heatmap of the cytotoxicity of plant extracts against two cell lines (A549 and WI-26 VA4); (B) Plant extracts with pronounced cytotoxic effects against A549 and WI-26 VA4 cell lines; (C) Cytotoxicity of plant extracts in different solvents. Legend. H: Hexane; A: Acetone; E: Ethanol; M: Methanol; W: Water.

by a heatmap clustering ("Fig 6A") and the inhibition values along with their standard deviation are listed in "S8 and S9 Tables in S1 File." Interestingly, only a few extracts showed strong inhibition against the cell lines, while the majority of extracts had low or no-cytotoxicity at the concentration tested. In total, 10 extracts were active (IV > 50%) against A549 and 9 extracts against WI-26 VA4, while 5 extracts were more active (IV > 70%) against A549 and WI-26 VA4 ("Fig 6B"). The ethanol and methanol extracts of *A. indica* showed strong activity against A549 (90% and 90%, respectively) and WI-26 VA4 (74% and 79%, respectively) ("Fig 6C"). The methanol extract of *S. moorcroftiana* only showed inhibition

(69%) against cancer cells but not against noncancerous cells, which suggests the presence of compounds with potential anti-cancer activity. The methanol extracts were more active (40%) against cancer cells, while hexane extract showed higher activity (44%) against noncancer cells compared with other extractant solvents. The Chi-square test results revealed that there is a significant association between the solvents used and the number of cytotoxic extracts against both A549 (p-value < 0.001) and WI 26 VA4 (p-value < 0.001) cell lines. Based on the effects of plant extracts on tumoral and normal cell lines to provide information on the safety of these medicinal plants, we found that most of the active plants have no significant cytotoxicity, which supports their longstanding use by the local population. Only a few extracts showed activity against cancer cells, which could be studied further for isolating anti-cancer compounds.

Cytotoxicity effects have been reported for outer bark of *J. regia* against MDA-MB-231 and A549 cell lines [125], peel extract of *P. granatum* against HTB140, HTB177, MCF7, HCT116 human cancer cell lines, MRC-5 normal fibroblasts [126] as well as against HeLa and REF cell lines [127], methanolic extracts of *A. maritima* [31], crude extracts of *B. lyceum* against HepG2 cells [128], the hydroalcoholic extract of *N. officinale* against the HeLa cell line [129], various extracts of *D. ambrosioide* against the GM 07492 cell lines and normal human fibroblasts [130], crude extracts of *D. salicifolia* against the MCF-7 cancer cell [131], extracts of *A. indica* against the MCF-7 breast cancer cell line [132], *Z. Oxyphylla* [133] and essential oils of *T. linearis* against two breast cancer cell lines MCF-7 and T47D [134], different extracts of *M. longifolia* against the MCF-7 cell line [135], extracts of *C. procera* against hepatoma (Huh7) and non-hepatoma (COS-1) cell lines and non-transformed hepatocytes (AML12) [136], crude essential oils extract of *O. vulgare* against HepG2 [137], essential oils and extracts of *M. Spicata* against Vero, Hela and HEp-2 cell lines [138], water extracts of *P. major* against human leukemia, lymphoma and carcinoma cells [139], and methanolic fraction of A. *bracteosa* against MCF-7 and HEp-2 tumour cell lines [140]. However, we could not find reports on the cytotoxicity effects of *S. moorcroftiana* against any cancerous or noncancerous cell lines, and majority of the test extracts were also not reported against the used cell lines. Our findings with cytotoxicity tests are in line with the available literature, which confirms the safe traditional use of these plants. In conclusion, except for the extracts of *A. indica* and *S. moorcroftiana*, all other extracts are mostly safe, and attractive for further studies. The identification and isolation of bioactive compounds from the studied plants with potential against the reported susceptible pathogens and exploration of their mechanism of action, structure activities relationship, in silico identification of the active compounds, and schematic classification of molecules per activity are recommended for future studies.

## Conclusion

The current study summarizes the ethnomedicinal documentation, phytochemical characterization, TLC profiling, antibacterial, antifungal, antibiofilm, and cytotoxicity effects of select traditional medicinal plants of the Swat region of Pakistan. Together with the available literature, it supports their safe human medical use. This study confirms the importance of indigenous knowledge of the local population and their use of medicinal plants for various remedies in selecting plants for drug discovery. Further work is needed to isolate the bioactive compounds from the most promising plants, and to explore their structure activities relationship, in silico identification of the active compounds, schematic classification per activity, and mechanism of action for discovering new drug candidates.

## Supporting information

**S1 File.** Supplementary tables for ethnomedicinal documentation, phytochemical characterization, and antimicrobial, antibiofilm, and cytotoxicity assays. **S1 Table.** Survey form for the data and sample collection of traditional medicinal plants from Swat, Pakistan. **S2 Table.** Quantitative analysis of the proportional distribution of each phytochemical constituent across 17 traditional medicinal plants from Swat, Pakistan. **S3 Table.** Antimicrobial activities (percent inhibition values of two replicate experiments, mean ± SD) of plant extracts against gram-positive bacteria. **S4 Table.** Antimicrobial activities

(percent inhibition values of two replicate experiments, mean ± SD) of plant extracts against gram-negative bacteria. **S5 Table.** Antimicrobial activities (percent inhibition values of two replicate experiments, mean ± SD) of plant extracts against fungi. **S6 Table.** Antibiofilm activity (percent growth inhibition of two replicate experiments, mean ± SD) of plant extracts against *Staphylococcus aureus* (USA 300). **S7 Table.** Antibiofilm activity (percent growth inhibition of two replicate experiments, mean ± SD) of plant extracts against *Candida albicans* (SC 5314). **S8 Table.** Cytotoxicity (cell viability inhibition of two replicate experiments, mean ± SD) of plant extracts against A549 cell lines. **S9 Table.** Cytotoxicity (cell viability inhibition of two replicate experiments, mean ± SD) of plant extracts against WI-26 VA4 cell lines. Legend for S3–S9 Tables: No. 1–17 = Plant samples; sub-numbers (−1 to −5) = solvents used (hexane, acetone, ethanol, methanol, water); E1 = Experiment 1; E2 = Experiment 2; M ± SD = Mean ± Standard Deviation.
(PDF)

**S2 File.**   Raw data corresponding to Figs 2–6 and selected tables. **S1 Data.** Raw data for Fig 2A. **S2 Data.** Raw data for Fig 2B. **S3 Data.** Raw data for Fig 2C. **S4 Data.** Raw data for Fig 3. **S5 Data.** Raw data for Fig 4A. **S6 Data.** Raw data for Fig 4B. **S7 Data.** Raw data for Fig 4C. **S8 Data.** Raw data for Fig 5A. **S9 Data.** Raw data for Fig 5B. **S10 Data.** Raw data for Fig 5C. **S11 Data.** Raw data for Fig 6A. **S12 Data.** Raw data for Fig 6B. **S13 Data.** Raw data for Fig 6C. **S14 Data.** Photographs of the plant samples for phytochemical analysis (Table 2). **S15 Data**. Photographic images of TLC fingerprints corresponding to Table 3.
(PDF)

**S3 File.  Questionnaire on Inclusivity in Global Research.** Contains responses to PLOS ONE's inclusivity survey addressing research design, authorship, and demographic representation.
(PDF)

## Acknowledgments

The authors are thankful to the authorities of University of Swat Pakistan and KU Leuven Belgium for providing facilities for conducting this research work. The author, Ajmal Khan, is deeply grateful to the Higher Education Commission of Pakistan, the University of Swat, Pakistan, and KU Leuven, Belgium, for their valuable support and facilitation during his PhD studies. The authors are highly grateful to Dr. Zahid Ullah, Assistant Professor, Centre for Plant Science and Biodiversity, University of Swat, Pakistan for identification of plants. The authors are also grateful to all the local traditional knowledge holders especially Prof. Mehboob-ur-Rahman (alias Khuban) of Matta, Swat, and to all other individuals and institutions who contributed directly or indirectly to this research study.

## Author contributions

**Conceptualization:** Ajmal Khan, Sujogya Kumar Panda, Liliane Schoofs, Walter Luyten.

**Data curation:** Ajmal Khan, Sujogya Kumar Panda, Haibo Hu, Liliane Schoofs, Walter Luyten.

**Formal analysis:** Ajmal Khan, Sujogya Kumar Panda.

**Funding acquisition:** Ajmal Khan, Liliane Schoofs, Walter Luyten.

**Investigation:** Ajmal Khan, Sujogya Kumar Panda, Haibo Hu, Walter Luyten.

**Methodology:** Ajmal Khan, Sujogya Kumar Panda, Haibo Hu, Walter Luyten.

**Project administration:** Liliane Schoofs, Walter Luyten.

**Resources:** Liliane Schoofs, Walter Luyten.

**Software:** Liliane Schoofs, Walter Luyten.

**Supervision:** Sujogya Kumar Panda, Liliane Schoofs, Walter Luyten.

**Validation:** Ajmal Khan, Sujogya Kumar Panda, Haibo Hu.

**Visualization:** Ajmal Khan, Sujogya Kumar Panda, Haibo Hu.

**Writing – original draft:** Ajmal Khan.

**Writing – review & editing:** Ajmal Khan, Sujogya Kumar Panda, Haibo Hu, Liliane Schoofs, Walter Luyten.

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
