## [Decision Letter · Decision Letter 0]

28 Mar 2025

Dear Dr. Khan,

Thank you for submitting your manuscript to PLOS ONE. After careful consideration, we feel that it has merit but does not fully meet PLOS ONE’s publication criteria as it currently stands. Therefore, we invite you to submit a revised version of the manuscript that addresses the points raised during the review process.

We look forward to receiving your revised manuscript.

Kind regards,

Ghulam Yaseen, Ph.D.

Academic Editor

PLOS ONE

Journal Requirements:

2. Please include the following request in the decision letter, and ping me with follow up. “Please include a complete copy of PLOS’ questionnaire on inclusivity in global research in your revised manuscript. Our policy for research in this area aims to improve transparency in the reporting of research performed outside of researchers’ own country or community. The policy applies to researchers who have travelled to a different country to conduct research, research with Indigenous populations or their lands, and research on cultural artefacts. The questionnaire can also be requested at the journal’s discretion for any other submissions, even if these conditions are not met.  Please find more information on the policy and a link to download a blank copy of the questionnaire here: https://journals.plos.org/plosone/s/best-practices-in-research-reporting. Please upload a completed version of your questionnaire as Supporting Information when you resubmit your manuscript.

Additional Editor Comments:

Comments from PLOS Editorial Office: We note that one or more reviewers has recommended that you cite specific previously published works. As always, we recommend that you please review and evaluate the requested works to determine whether they are relevant and should be cited. It is not a requirement to cite these works. We appreciate your attention to this request.

Reviewers' comments:

Reviewer's Responses to Questions

**Comments to the Author**

1. Is the manuscript technically sound, and do the data support the conclusions?

Reviewer #1: Partly

Reviewer #2: Yes

Reviewer #3: Yes

Reviewer #4: Yes

2. Has the statistical analysis been performed appropriately and rigorously?

Reviewer #1: No

Reviewer #2: N/A

Reviewer #3: Yes

Reviewer #4: Yes

3. Have the authors made all data underlying the findings in their manuscript fully available?

Reviewer #1: No

Reviewer #2: Yes

Reviewer #3: Yes

Reviewer #4: No

4. Is the manuscript presented in an intelligible fashion and written in standard English?

Reviewer #1: Yes

Reviewer #2: Yes

Reviewer #3: Yes

Reviewer #4: Yes

Reviewer #1: 1- Why, in cytotoxic assays, has IC50 (50% viable cells) not been calculated?

2- To compare the results, one-way ANOVA with post-test was used?

3- On page 20, line 540 the authors describe "S. aureus, S. epidermidis, E. coli, K. 540 pneumoniae, P. aeruginosa, C. albicans and A FEW OTHERS HAVE BEEN REPORTED." Avoid using inaccurate information.

4- Data were not fully available from the underlying manuscript. The authors have presented the mean ± SD.

Reviewer #2: The study explores ethnomedicinal documentation, phytochemical characterization, and antibacterial, antifungal, and antibiofilm effects of traditional medicinal plants in the Swat region of Pakistan, highlighting the importance of indigenous knowledge in drug discovery. A very basic way of finding certain targets in many samples did imparted something to science.

Reviewer #3: First of all, I would like to congratulate with the authors for this very nice work: A WELL-DONE JOB!!!

1. Please, make the title shorter, EXAMPLE: "Chemical and Biological Characterization of Medicinal Plants Prom Pakistan"

2. You are providing really a lot of data: BUT, I do not see classification tables of molecules per activities.

Please, referee to this article as example: "Fuoco, Domenico. "Classification framework and chemical biology of tetracycline-structure-based drugs." Antibiotics 1.1 (2012): 1-13."

Specifically, I am referring to the charts named Scheme 1 and Scheme 2.

Congratulations again for this very nice job.

Reviewer #4: The authors described 17 traditional medicinal plants from multiple aspects and assessed the biological activities of their extracts. One major limitation is the authors did not test the detailed chemical composition of these extracts. The following are my comments:

1. This paper mentioned the multiple biological activities of medicinal plants, authors should describe the antimicrobial, antiviral, anticancer, and other effects of the medicinal plants or extracted compounds in more detail. I strongly recommend the following publications (PMID): 27721399, 28117389, 29770277, 35150482, 35355727, 37264383, and 38983788.

2. The authors should show pictures of the ecology and medicinal parts of all these 17 medicinal plants.

3. The authors mentioned that the extracts detected several phytochemical compounds such as alkaloids, flavonoids, phenols, steroids, terpenoids, coumarins, tannins, saponins, chalcones, and quinones. What’s the ratio of these compounds in these 17 plants and 85 extracts?

4. These extracts contain hundreds or thousands of compounds, the manuscript lack a description of their chemical components.

5. I suggest screening some compounds with novel structures for various biological activity tests.

**Do you want your identity to be public for this peer review?** For information about this choice, including consent withdrawal, please see our Privacy Policy

Reviewer #1: No

Reviewer #2: No

Reviewer #3: **Yes: ** DOMENICO FUOCO

Reviewer #4: No

---

## [Author Response · Author response to Decision Letter 1]

12 Jun 2025

June 12, 2025

The Academic Editor,

PLOS ONE.

Subject: Submission of Revised Manuscript

Dear Prof. Dr. Ghulam Yaseen,

We are grateful to you and the reviewers for your constructive feedback and fruitful suggestions on our manuscript titled ‘‘Ethnomedicinal documentation, phytochemical characterization, and biological evaluation of the traditional medicinal plants from Swat region of Pakistan’’ by Ajmal Khan, Sujogya Kumar Panda, Haibo Hu, Liliane Schoofs, and Walter Luyten, to be reconsidered for publication as an original article in the PLOS ONE.

We are thankful for your gentle decision letter and for giving us the chance to revise and improve our manuscript. We are highly indebted to you, the editorial office and the reviewers for providing valuable feedback, which has improved the quality of our work.

We have carefully addressed each point raised by the reviewers and the editorial office and have revised the manuscript after incorporating their comments and suggestions to the best of our level. We have ensured full compliance with PLOS ONE’s formatting guidelines, ethics statement placement, financial disclosure statement, and the submission of the global research inclusivity questionnaire.

We have highlighted all the changes in the revised manuscript as red (Reviewer 1), blue (Reviewer 3), orange (Reviewer 3&4), purple (Reviewer 4), and dark red (Editorial office remarks or PLOS ONE’s guidelines or changes by co-authors for further improvement of the quality of the manuscript). We have also added details of comments/suggestions and the revised changes/ rebuttal comments in the section below ‘‘Responses to Reviewers’’.

Responses to Reviewers:

Reviewer #1:

We sincerely appreciate the valuable feedback of the Reviewer 1, which has greatly improved the quality of our work and we have addressed his comments in the revised manuscript (highlighted as “red’’), and the detailed explanations are provided below.

1- The present study focuses on the preliminary screening of the traditional medicinal plants of Swat region of Pakistan to explore their qualitative phytochemical characteristics along with the antimicrobial and antibiofilm potential. The cytotoxicity assays were used as a screening tool to identify non-cytotoxic plant extracts for further therapeutic evaluation. Therefore, the IC50 Values of the cytotoxicity assays were not calculated.

2- We appreciate the reviewer’s suggestion to perform one-way ANOVA test, but due to the nature of our data and for conveying our message more clearly for our readers we employed Tukey’s HSD and Chi-Square tests as an alternative tests in the revised manuscript.

3- The provided information have been updated accordingly.

4- The full data of all the experiments along with mean ± SD has been included in the revised manuscript with the addition of four detailed tables as supplementary material.

Reviewer #2:

The acknowledging remarks of the Reviewer 2 regarding the value of our research are highly appreciated.

Reviewer #3:

We are thankful to Reviewer 3 for his encouraging words and positive feedback. We have addressed his comments in the revised manuscript (highlighted in blue) and the detailed explanations are given below.

1. The title of the manuscript has been revised accordingly.

2. We appreciate the suggestion of the reviewer regarding inclusion of schematic classification of molecules per activity and structure activity relationship. Although, such detailed analyses are beyond the scope of our current study, but after getting reviewer’s suggestion, we have added a section (orange color) which summarizes the major chemical constituents and associated bioactivities of the studied plants. Moreover, we have recommended the structure activity relationship with the relevant suggested publications for future research as highlighted in the revised manuscript.

Reviewer #4: We are thankful to the Reviewer 4 for his productive comments and significant suggestions, which has improved the quality of our work. We have addressed his comments in the revised manuscript (highlighted in purple), and explained each point below.

We appreciate the observation of the reviewer regarding the detailed chemical composition of the plant extracts. We want to clarify that our main objective of this study was ethnomedicinal documentation of the traditional medicinal plants of Swat region of Pakistan, followed by preliminary phytochemical analysis to identify the major classes of bioactive compounds like alkaloids, flavonoids, saponins, etc. We agree that the detailed chemical profiling using LCMS-MS or NMR is fruitful, however, it’s beyond the scope and current resources of our study. However, we do intend to perform advanced chemical characterization of these plants in the near future and have also recommended it as important direction for other researchers.

1. We are thankful to the reviewer 4 for providing valuable references. We have included a more detailed overview of the antimicrobial, antibiofilm, and cytotoxicity activities of the studied plants in Supplementary material. We have included cytotoxicity activities just to select the non-cytotoxic extracts for our future studies. However, anticancer and antiviral activities was not the focus of this study, and our current resources and facilities also do not support such activities. We have already discussed the activities of the plants and have recommended the suggested relevant publications (27721399, 28117389, 29770277, 35355727, and 38983788) regarding molecular analyses of the extracts and in silico identification of the bioactive compounds of medicinal plants for future research. It will definitely help us and other researchers in future, but unfortunately our current resources and limitations do not allow us to include such a data as it is beyond scope of this work. We are grateful to the reviewer for his overall fruitful recommendations which will help us and other researchers in future.

2. The pictures of all the 17 medicinal plants have been included in the revised manuscript in a new figure (“Fig 1.tif”).

3. The quantitative analysis of the proportional distribution of the detected classes of phytochemical compounds such as alkaloids, flavonoids, phenols, steroids, terpenoids, coumarins, tannins, saponins, chalcones, and quinones of all the medicinal plants have been included in the revised manuscript as supporting information (‘‘S2 Table”).

4. We are thankful to the reviewer for suggesting description of the chemical components of the medicinal plants. Although, such detailed analyses are beyond the scope of our current study, but after getting reviewer’s suggestion, we have added a literature-based section (orange color) which summarizes the major chemical constituents and associated bioactivities of the studied plants and have recommended comprehensive analyses too.

5. We highly appreciate the suggestion of the reviewer regarding screening of novel compounds for various biological activities. Although, as stated earlier, such detailed characterization is beyond the scope of our current study but we intend to do the screening of novel compounds for diverse biological activities in the future which is the main target of our project. We are obliged for this valuable suggestion and will certainly keep it in mind for our ongoing and future research projects.

We are thankful to you once again for your valuable feedback. We believe the changes in the revised version have improved the manuscript, and we hope it now meets your expectations. We look forward to your positive feedback.

Warm Regards,

Ajmal Khan

Department of Biology, Animal Physiology and Neurobiology Section,

KU Leuven, 3000 Leuven, Belgium / Centre for Animal Sciences and Fisheries, University of

Swat, Pakistan.

Email: ajmal.khan@kuleuven.be, ajmalkhan399@hotmail.com

Phone: +32492957909 / +923439583481

---

## [Decision Letter · Decision Letter 1]

22 Jul 2025

Ethnomedicinal documentation, phytochemical characterization, and biological evaluation of the traditional medicinal plants from Swat region of Pakistan

PONE-D-24-44242R1

Dear Dr. Khan,

We’re pleased to inform you that your manuscript has been judged scientifically suitable for publication and will be formally accepted for publication once it meets all outstanding technical requirements.

Kind regards,

Ghulam Yaseen, Ph.D.

Academic Editor

PLOS ONE

Additional Editor Comments (optional):

Reviewers' comments:

Reviewer's Responses to Questions

**Comments to the Author**

Reviewer #3: All comments have been addressed

2. Is the manuscript technically sound, and do the data support the conclusions?

Reviewer #3: Yes

3. Has the statistical analysis been performed appropriately and rigorously?

Reviewer #3: N/A

4. Have the authors made all data underlying the findings in their manuscript fully available?

Reviewer #3: Yes

5. Is the manuscript presented in an intelligible fashion and written in standard English?

Reviewer #3: Yes

Reviewer #3: I was among the first pool of reviewers to get this article at its original submission. At time, i endorsed the publication of this work. I am happy to see that authors have continued to work on this paper and they have been to address the concerns of the other reviewers. Congratulation of this excellent job.

**Do you want your identity to be public for this peer review?** For information about this choice, including consent withdrawal, please see our Privacy Policy

Reviewer #3: **Yes: ** DOMENICO FUOCO
